# Dual membrane receptor degradation via folate receptor targeting chimera

Zhen Wang [1,11], Zhixin Li[2,3], Jenny Högström [4], Hiroyuki Inuzuka[1], Rui Jing[5], Peiqiang Yan[1], Tao Hou[1], Yihang Qi [1], Daoyuan Huang[1], Jingchao Wang[1], Ting Wu[6,7,8], Xiaoying Shi [9], Bolin Liu [10], Taru Muranen [4], Dingpeng Zhang [1,11] ✉ & Wenyi Wei [1] ✉

Cancer drug resistance poses a significant challenge in oncology, often driven by intricate cross-talk among membrane-bound receptors that compromise mono-targeted therapies. We develop a dual membrane receptor degradation strategy leveraging Folate Receptor α (FRα) to address this issue. Folate Receptor α Targeting Chimeras-dual (FolTAC-dual) are engineered degraders designed to selectively and simultaneously degrade distinct receptor pairs: (1) EGFR/HER2 and (2) PD-L1/VISTA. Through modular optimization of modality configurations and geometries, we identify the "string" format as the most effective construct. Mechanistic studies demonstrate an ~85% increase in EGFR-binding affinity compared to the conventional knob-into-hole design, likely contributing to the improved efficiency of dual-target degradation. Proof-of-concept studies reveal that EGFR and HER2 FolTAC-dual effectively counteracts resistance in Trastuzumab/Lapatinib-resistant HER2-positive breast cancer models, while PD-L1 and VISTA FolTAC-dual rejuvenates immune responses in PD-L1 antibody-resistant syngeneic mouse models. These findings establish FolTAC-dual as a promising dual-degradation platform for clinical translation.

Cancer drug resistance is a persistent barrier to the success of targeted therapies, often arising from compensatory signaling between membrane-bound receptors that bypass mono-targeted inhibition[1–4]. For instance, resistance to HER2-targeted therapies like Trastuzumab frequently involves upregulated EGFR or HER3 signaling[5–10], while immunotherapy resistance is driven by alternative immune checkpoints, such as VISTA, compensating for PD-L1 blockade[11–15]. These mechanisms highlight the pressing need for innovative multi-receptor targeting approaches that mitigate redundancy while maintaining

specificity. Bispecific antibodies have been developed to block two receptor-mediated pathways[16]. However, the resistance problem remains a potential hurdle for this strategy[17]. These limitations highlight the need for alternative approaches—specifically, dual-targeted therapeutic platforms that eliminate, rather than merely inhibit, key membrane receptors to more effectively overcome resistance mechanisms.

Emerging technologies such as LYTACs[18] or LYTAC-like platforms[19] have demonstrated the potential for targeted membrane

[1]Department of Pathology, Beth Israel Deaconess Medical Center, Harvard Medical School, Boston, MA, USA. [2]Department of Medical Oncology, Dana-Farber Cancer Institute, Boston, MA, USA. [3]Broad Institute of Massachusetts Institute of Technology (MIT) and Harvard University, Cambridge, MA, USA. [4]Department of Medicine, Beth Israel Deaconess Medical Center, Harvard Medical School, Boston, MA, USA. [5]Houston Methodist Cancer Center/Weill Cornell Medicine, Houston, TX, USA. [6]Division of Hematology/Oncology, Boston Children's Hospital, Boston, MA, USA. [7]Department of Pediatric Oncology, Dana-Farber Cancer Institute, Boston, MA, USA. [8]Department of Pediatrics, Harvard Medical School, Boston, MA, USA. [9]Channing Division of Network Medicine, Brigham and Women's Hospital, Boston, MA, USA. [10]Department of Interdisciplinary Oncology, Stanley S. Scott Cancer Center, LSU Health Sciences Center, New Orleans, LA, USA. [11]These authors contributed equally: Zhen Wang, Dingpeng Zhang. ✉e-mail: dzhang13@bidmc.harvard.edu; wwei2@bidmc.harvard.edu

protein degradation. The LYTAC is developed based on the cation-independent mannose-6-phosphate receptor, a pioneering membrane degrader, and provides a paradigm for protein degradation. Subsequently, various receptors such as integrin[20], RNF43[21], TfR1[22], and FR[23] have been successfully developed as effectors of membrane protein degradation. However, these approaches face limitations, such as restricted adaptability for simultaneous multi-target degradation of membrane proteins and limited receptor exploration across diverse cancer types[18,21,24–26]. To address these limitations, dual membrane protein degradation platforms based on polymer modalities, such as AptLYTACs, have been introduced[27]. While promising, these systems have notable drawbacks, including weaker targeting specificity, a narrower range of targetable proteins, and significant challenges in drug delivery and development[27].

In this context, Folate receptor α (FRα) is a particularly compelling candidate due to its role in endocytosis[28–31], driven by high-affinity folate binding ($K_D = 10$ pM–1 nM)[28,29]. In addition, FRα is critical for the survival of dividing cells as it supports the synthesis of essential biomolecules, including amino acids and nucleic acids[30,31]. Its high expression in various cancers, along with a proven safety profile, makes FRα an attractive target for therapeutic applications. Importantly, FRα's safety and efficacy have been validated through the recent FDA approval of mirvetuximab soravtansine-gynx for FRα-positive, platinum-resistant epithelial ovarian, fallopian tube, or primary peritoneal cancers[32,33]. The established safety profile, combined with its high expression in diverse cancers, positions FRα as an ideal receptor to harness for targeted protein degradation strategies.

Motivated by the clinical relevance and endocytic properties of FRα, we introduce the Folate Receptor α Targeting Chimeras-dual degrader platform (FolTAC-dual), a rationally modular design that enables the simultaneous degradation of two membrane proteins, offering a versatile strategy to overcome drug resistance (Fig. 1a, b). Built on an antibody scaffold, FolTAC-dual integrates optimized receptor-targeting binders, linker architectures, binding configurations, and degrader geometries to co-target EGFR and HER2 as a representative example. Notably, we developed a distinct dual degrader format-referred to as the "string" configuration-which exhibited notably enhanced degradation efficiency of EGFR and HER2 compared to the conventional knob-into-hole design. In vitro analysis reveals that the string version enhances the binding affinity of the EGFR-targeting arm by 85%, underscoring the synergistic effect of co-binding in achieving superior degradation performance. Our in vitro and in vivo studies demonstrate that the EGFR and HER2 FolTAC-dual degrader effectively suppresses tumor growth in Trastuzumab/Lapatinib-resistant HER2-positive breast cancer (HER2 + BC) models[34,35]. Moreover, we extend this approach to target and degrade immune checkpoint molecules PD-L1 and VISTA, further highlighting the versatility and therapeutic potential of FolTAC-dual technology. The PD-L1 and VISTA-targeted FolTAC-dual degrader is shown to enhance immune activation in an engineered PD-L1 antibody-resistant MC38-C57BL/6 syngeneic mouse model, underscoring its ability to modulate the tumor microenvironment (TME)[36–39]. Collectively, these findings establish a robust framework for leveraging receptor degradation to combat drug resistance and position the FolTAC-dual platform as a promising therapeutic modality in cancer treatment.

In this work, we systematically engineer and evaluate the FolTAC-dual platform by applying it to two clinically relevant target pairs—EGFR/HER2 and PD-L1/VISTA—demonstrating its ability to overcome therapeutic resistance in both oncogenic and immunosuppressive contexts. Through comparative mechanistic studies, we show that the optimized "string" configuration enhances binding affinity and degradation efficacy, leading to improved therapeutic outcomes in resistant cancer models.

## Results

### Rational design of FolTACs-dual

The membrane proteins often work synergistically to drive tumor proliferation and survival, contributing to resistance against cancer therapies (Supplementary Fig. 1a). For example, the EGFR and HER2 signaling pathways amplify downstream phosphorylation[8], while the B7-H1/PD-L1 immune checkpoints facilitate immune evasion[40]. These synergistic mechanisms frequently lead to resistance against single-agent therapies, such as PD-L1 antibody blockade[41,42] or Trastuzumab treatment for HER2-positive breast cancer (HER2 + BC). Although bispecific antibodies represent a significant advancement, their efficacy is often limited by persistent resistance, necessitating further development[43,44]. To overcome these challenges, we developed Fol-TAC-dual, a class of degraders to simultaneously eliminate two Proteins of Interest (POI) rather than inhibit their activity (Fig. 1a, b). The versatility of the FolTAC platform was demonstrated through the development of multiple constructs designed to target and degrade diverse membrane proteins with distinct functions (Supplementary Fig. 1b–g), thereby showcasing its potential for broad applications in cancer therapy.

### Folate receptor α is highly expressed in various cancer subtypes

FRα is known to be highly expressed in gynecological, breast, and lung cancers[45]. However, there is still a lack of comprehensive analysis of its expression profile across a broader range of cancer types. To address this gap and assess the practical applications of the FolTAC technology, we evaluated FRα expression levels in various tumor tissues. Using flow cytometry, we measured FRα levels on the cell surface of both cancerous and non-cancerous cell lines derived from multiple tissue types. Our results revealed that FRα is prominently expressed across various cancer types compared to normal cells (Fig. 1c and Supplementary Fig. 1h–j). Complementary transcriptomic analyses from the Genotype-Tissue Expression (GTEx) and Cancer Genome Atlas (TCGA) public databases further revealed that *FOLR1*, the gene encoding FRα, is overexpressed in numerous cancers relative to normal tissues (Supplementary Fig. 1h). Notably, specific subtypes of gynecologic, breast, lung, and other cancers demonstrate relatively high expression levels of FRα (Supplementary Fig. 1i, j). These findings were strongly corroborated by the flow cytometry data, showing a consistent pattern of elevated FRα levels in cancer cells compared to normal counterparts. This comprehensive characterization of FRα expression highlights its potential as a therapeutic target and informs the subsequent design, development, and application of the FolTAC technology in selected cancer subtypes.

### Engineering FolTAC for single degradation, poised for a dual strategy

Our FolTAC system leverages rational engineering of antibody scaffolds, including the optimization of binding ligands, linkers, and isotypes, to achieve efficient dual degradation (Fig. 1d, e). We hypothesized that the effectiveness of FolTAC-dual is influenced by several critical factors. First, we compared the HER2 targeting binder between Trastuzumab Single-Chain Fragment Variable (scFv) and HER2 affibody, revealing that the affibody binder showed relatively greater degradation efficiency (Fig. 1d). Next, we explored the role of the Cathepsin B (CatB) linker, previously known to be critical in TransTAC degraders[26]. Upon evaluation, incorporation of the CatB linker reduced the degradation efficiency in HER2 and EGFR FolTACs (Fig. 1d and Supplementary Fig. 2a–c) and even caused hook effects[46] for the TfR1 FolTAC-mediated degradation (Supplementary Fig. 2b). This phenomenon may be in part attributed to the role of the effector FRα in promoting lysosomal trafficking, which facilitates the retention and degradation of targeted proteins, a process potentially disrupted by the addition of the CatB linker (Supplementary Fig. 2a–c). In addition, we tested various isotypes of EGFR FolTACs—v0.2, 0.4, and 0.5—

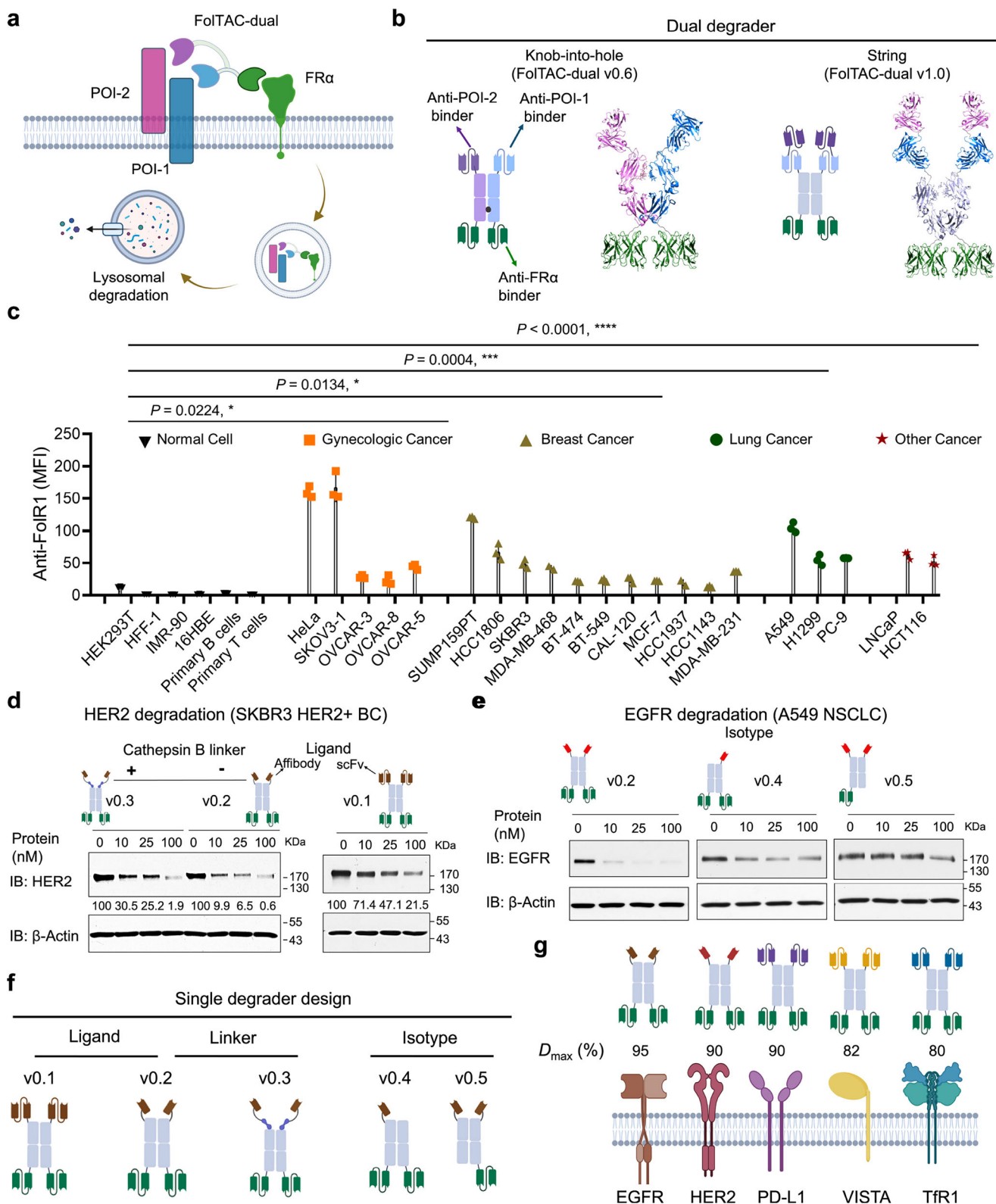

and identified v0.2, featuring two binding arms, as the most effective configuration (Fig. 1e, f). To evaluate the degradation efficiency of FolTAC v0.2 on single targets in cancers and immune-oncology, we engineered various FolTACs to target and degrade TfR1, EGFR, HER2, PD-L1, and VISTA (Fig. 1g and Supplementary Fig. 2d, e). Our first target is TfR1, a key receptor for iron import that is highly expressed in various tumor types[47]. The TfR1 FolTAC was developed by fusing the TfR1 scFv binder H7[48] to the Farletuzumab scFv[49], meanwhile, the

H7-FC[48] was used as a control. Treatment with H7-FC caused minimal TfR1 degradation, whereas the TfR1 FolTAC exhibited dose-dependent degradation efficiency, achieving 50–80% degradation (Supplementary Fig. 2f). Similarly, we developed the EGFR FolTAC and achieved >95% degradation efficiency in 10–100 nM concentrations using PC9 cell lines[50–52] (Supplementary Fig. 2g). Like the EGFR FolTAC developed above, the HER2 FolTAC was developed following the same principle by fusing HER2 targeting affibody[53–55] to the FRα targeting scFv-

**Fig. 1 | Overview of the FolTAC technology and Folate Receptor α (FRα) expression analysis. a** A schematic illustration of FolTAC-mediated endocytosis of two POIs. FolTAC induces proximity of FRα and a POI at the cell surface, leading to co-internalization of the complex to the endosome/lysosome for degradation. **b** Schematics and structures of the knob-into-hole and the string FolTAC-dual geometry. **c** Relative cell surface FRα expression levels across various non-tumorigenic and cancer cell lines characterized by flow cytometry. Cell lines were grouped into five categories: normal (black downward-pointing triangles), gynecologic cancer (orange squares), breast cancer (olive-green upward-pointing triangles), lung cancer (green circles), and other cancers (red stars). Normal cells (HEK293T, HFF-1, IMR-90, h-8LD, Primary B cells, and Primary T cells) served as the control group for statistical comparisons. Each data point represents the mean fluorescence intensity (MFI) of FRα staining. Data are presented as mean ± standard deviation (SD), with $n = 3$ biological replicates per condition. The quantification of the data was analyzed using two-sided Student's $t$-tests using GraphPad Prism software, and exact $p$-values are shown above the figure bars. Differences were considered statistically significant at $p < 0.05$. *, ***, ****: $p < 0.05$, $p < 0.001$, $p < 0.0001$. **d** Western blotting of HER2 degradation by HER2 Trastuzumab (Traz) FolTAC v0.1, HER2 affibody without Cathepsin B linker FolTAC v0.2, and HER2 affibody with Cathepsin B linker FolTAC v0.3 in SKBR3 cells. Relative protein levels were labeled with numbers below the indicated bands. **e** Western blotting of EGFR degradation by various EGFR isotypes v0.2, v0.4, and v0.5 in A549 cells. Relative protein levels were labeled with numbers below the indicated bands. **f** Schematics of the evolution of FolTAC by rational design, including the ligand, linker, and isotype optimization. **g** Maximal degradation efficiency ($D_{max}$, %) of FolTAC across various targets achieved through rational design. Created in BioRender. WANG, Z. (2025) https://BioRender.com/d5n9vj6.

Farletuzumab. With a concentration >25 nM, the degradation efficiency could be achieved at >90% (Supplementary Fig. 2h). To extend the FolTAC technology to immune checkpoints, we targeted programmed death-ligand 1 (PD-L1), a receptor that enables tumor cells to evade immune responses[56]. Treatment with the control did not cause the observable decrease of PD-L1 protein abundance, while SUM159PT cells treated with PD-L1 FolTAC resulted in > 90% degradation of the receptor starting in both 10 and 25 nM (Supplementary Fig. 2i). The last target for degradation is VISTA, an immune checkpoint molecule that interacts with PSGL-1 and is reported to be complementary to PD-L1[37,57,58]. OVCAR-5 ovarian cancer cells treated with the VSTB112 scFv-FC control showed no observable VISTA degradation, while the VISTA FolTAC-treated group demonstrated degradation and shift of the VISTA to a lower molecular weight band in a dose-dependent manner (Supplementary Fig. 2j). This could be possibly explained by the newly synthesized VISTA without the glycosylation modification upon the VISTA FolTAC-mediated degradation[59]. In addition, we treated cells with various FolTAC degraders targeting specific receptors and assessed FRα degradation (Supplementary Fig. 3a–c). We observed co-degradation events; however, the extent of FRα depletion varied depending on its baseline expression levels across different cancer cell lines. Notably, at concentrations of 25 nM or lower, cancer cell lines with high FRα expression (HeLa) exhibited minimal detectable reduction, whereas those with lower FRα levels (PC9) showed more pronounced depletion, suggesting that FolTAC activity may preferentially deplete FRα in contexts where its expression is limited. This highlights a potential trade-off between effective target engagement and unintended loss of FRα, depending on receptor expression levels in different cancer contexts. A time-course analysis of HER2 degradation is presented in Supplementary Fig. 3d. Collectively, these data demonstrate the broad applicability and effectiveness of the FolTAC technology for degrading diverse cell surface receptors in cancer.

While we detected the degradation of HER2, we also observed a significant inhibition of cell proliferation upon treatment with the HER2 FolTAC. This inhibitory effect may be attributed to disrupted HER2 downstream signaling. Therefore, to further investigate how FolTAC-mediated HER2 degradation affects downstream signaling, we focused on the PI3K/AKT/mTOR pathway as HER2 is known to activate key regulatory nodes crucial for breast cancer cell survival and proliferation[34,60,61]. Specifically, HER2 overexpression or hyperactivation drives cell growth by stimulating multiple signaling cascades, most notably the PI3K/AKT/mTOR axis, which in turn regulates various processes including metabolism, cell cycle progression, and protein synthesis[34,60,61]. To validate this hypothesis, we treated SKBR3 cells with HER2 FolTAC and examined AKT signaling markers by Western blot. Compared to controls, FolTAC treatment led to a marked reduction in the ratio of phosphorylated AKT at both S473 and T308 residues relative to total AKT, whereas only T308 phosphorylation was reduced in cells treated with the affibody-Fc control (Supplementary Fig. 3e–g).

We also observed moderate changes in p-mTOR levels (Supplementary Fig. 3e–g). Collectively, these findings suggest that FolTAC-mediated HER2 degradation primarily modulates key components of the PI3K/AKT pathway, with limited impact on mTOR signaling. In conclusion, the single-degradation optimization has provided key insights for our dual-degradation strategy. Moving forward, we aim to design the FolTAC-dual degrader by incorporating an scFv ligand, omitting the CatB linker, and employing the v0.2 isotype to maximize degradation efficiency.

## Engineering of FolTAC-dual to simultaneously degrade two membrane proteins

The design of membrane degraders has traditionally focused on targeting individual proteins. Recognizing the untapped potential of simultaneously degrading two membrane proteins—not achievable with bispecific antibodies—we developed the FolTAC-dual, a dual-target degrader based on the FolTAC technology.

Selecting an appropriate cancer cell line that expresses both target proteins is a critical consideration in developing FolTAC-dual technology for proof-of-concept demonstrations. Our RNA expression and western blot analysis of the co-expression profile revealed that the EGFR and HER2 pairs are present in some of the cancer cell lines, with PD-L1 and VISTA demonstrating a broad spectrum across various cell lines (Supplementary Fig. 4a–d). More importantly, these pairs have been identified as having complementary roles in oncological research, especially for gynecological, lung, and breast cancer cells[8,40].

From the above optimizations, we developed two distinct FolTAC-dual geometries: the "knob-into-hole" -v0.6 version and the "string" -v1.0 format (Fig. 2a). Both formats were evaluated in SKBR3 and PC9 cells, with the string version 1.0 exhibiting significantly enhanced EGFR and HER2 degradation efficiency compared to the knob-into-hole version 0.6. Control versions of both designs, which lacked the FRα scFv, exhibited minimal degradation activity, confirming the specificity of the FolTAC-dual system (Fig. 2b–e and Supplementary Fig. 4e). The second pair of targets investigated was PD-L1 and VISTA, two critical immune checkpoint molecules that suppress immune responses and enable tumor progression. The PD-L1/PD-1 axis is a well-established mechanism of immune evasion; however, only a subset of patients responds to PD-L1 inhibition[62–64]. One proposed mechanism for this resistance is the acidic tumor microenvironment, which enhances VISTA/PSGL-1 interactions and further inhibits phagocytosis (Supplementary Fig. 4f, g)[64–69]. We hypothesized that simultaneous degradation of PD-L1 and VISTA could likely produce synergistic anti-tumor effects. Testing this approach in SUM159PT and OVCAR-5 cells revealed that the string version (v1.0) was significantly more effective at degrading PD-L1 and VISTA compared to the knob-into-hole version (v0.6) and corresponding controls (Fig. 2f–i, Supplementary Fig. 4h, i). In addition, differential degradation levels were observed between VISTA and PD-L1 in Fig. 2g–i. One potential explanation is that differences in binding affinity and expression dynamics of the two POIs may

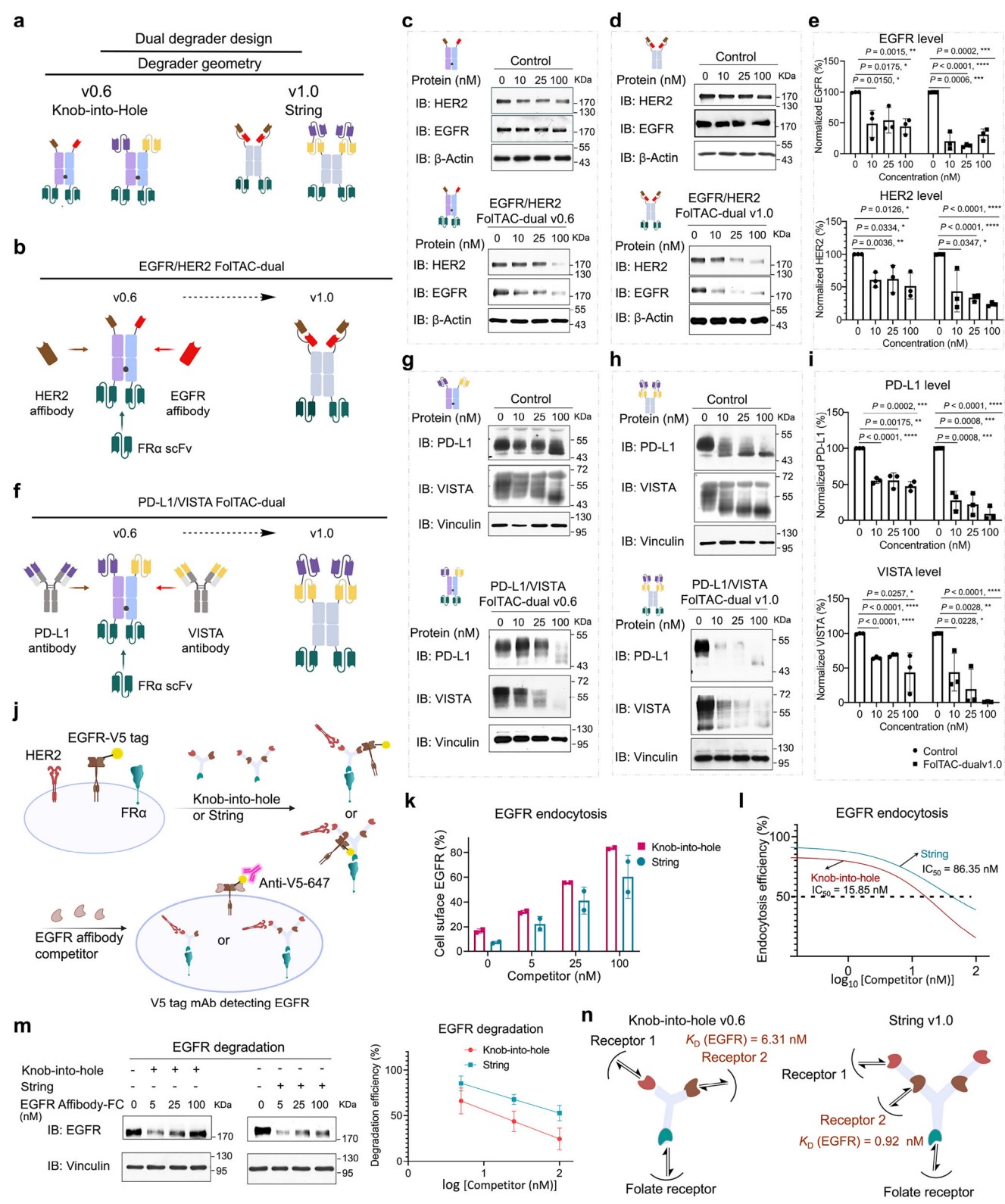

influence degradation efficiency. Notably, the appearance of neo-glycosylated or degraded forms of PD-L1 and VISTA (lower bands) adds further complexity, as these species may have differential accessibility to the degradation machinery or altered stability. These variations complicate accurate dose-to-degradation correlations and further suggest that substrate-specific factors, including post-translational modifications and receptor trafficking, may contribute to the observed differences in degradation levels. These observations provide valuable opportunities for further investigation and optimization in the engineering of functional modality design.

Together, these results establish the feasibility and practicality of the FolTAC-dual platform for dual receptor degradation. The platform's ability to degrade both the EGFR/HER2 and the PD-L1/VISTA pairs highlights its versatility and superior degradation efficiency, paving the way for future applications in dual-target inhibition and cancer therapy.

**Fig. 2 | Rational design and development of FolTAC-dual for simultaneous degradation of two membrane targets. a** Schematic illustrating the geometry-based evolution of FolTAC-dual. **b** EGFR/HER2 FolTAC-dual designs: knob-into-hole v0.6 and string v1.0 formats. **c–e** Western blot analysis of EGFR and HER2 degradation in SKBR3 cells by EGFR/HER2 FolTAC-dual, comparing knob-into-hole v0.6 (**c**) and string v1.0 (**d**), with quantification shown in (**e**). The EGFR/HER2 bispecific antibody lacking the FRα scFv served as the control. **f** Design schematics of PD-L1/VISTA FolTAC-dual in knob-into-hole v0.6 and string v1.0 formats. **g–i** Western blot analysis of PD-L1 and VISTA degradation in SUM159PT cells by PD-L1/VISTA FolTAC-dual, comparing knob-into-hole v0.6 (**g**) and string v1.0 (**h**), with quantification shown in (**i**). The control construct matched v1.0 but lacked the Farletuzumab arm. Samples and blots for Fig. 2e, i came from the same experiment and were processed in parallel. **j** Schematic of the competition binding assay to assess the relative binding affinity of knob-into-hole v0.6 and string v1.0. **k** EGFR receptor levels detected by flow cytometry using a V5-647 antibody after treatment with increasing

concentrations of EGFR affibody competitor. The analysis compared the internalization efficiency of the knob-into-hole (magenta squares) and string (cyan circles) formats. A conventional, untreated, or vehicle-only control group was not included, as the experiment was designed for direct comparison between the two formats. **l** IC$_{50}$ determination for EGFR engagement by knob-into-hole (red curve) and string (blue curve) formats. **m** Western blot analysis of EGFR levels following competition with affibody-Fc to evaluate degradation by knob-into-hole and string constructs. No untreated control was included to enable direct head-to-head comparison. **n** Schematic of geometry-influenced EGFR binding, with dissociation constants ($K_D$) calculated using the Cheng–Prusoff equation. Data in panels (**e**), (**i**), (**k**), (**l**), and (**m**) represent three independent biological replicates ($n = 3$) and are presented as mean ± SD. Statistical significance in panels 2e and 2i was assessed using two-sided Student's $t$-tests, and exact $p$-values are shown above the figure bars. $p < 0.05$ was considered significant. $p$ values: * <0.05, ** <0.01, *** <0.001, **** <0.0001. Created in BioRender. WANG, Z. (2025) https://BioRender.com/d5n9vj6.

## Mechanistic characterization of different FolTAC-dual geometries

To investigate the mechanism underlying the enhanced degradation efficiency of the string-version FolTAC-dual, we examined the dissociation kinetics ($K_D$) of the EGFR-binding arm as a representative component contributing to the overall degrader performance. We hypothesized that understanding the $K_D$ values would provide valuable insights into the differences between the string and knob-into-hole geometries. A competition binding assay was employed to quantify the $K_D$ of a specific EGFR binder for the EGFR receptor. Briefly, HEK cells were engineered to stably express V5-tagged EGFR, WT-HER2, and WT-FRα. The resulting cells were treated with either the string or knob-into-hole version of FolTAC-dual in the presence of increasing concentrations of an EGFR affibody competitor. Under these conditions, the endocytosis of only HER2 and FRα induced by FolTAC-dual continued unabated. However, the V5-tagged EGFR remained on the cell surface when its binding to the FolTAC-dual was competed out. The residual EGFR was then detected using an α-V5 tag conjugated to a fluorophore (647 nm) (Fig. 2j). As expected, increasing the concentrations of EGFR affibody competitor incrementally reduced EGFR endocytosis, as reflected by the higher percentage of EGFR retained on the cell surface (Fig. 2k). Our results indicated that the endocytosis of EGFR is independent of HER2, confirming the orthogonal targeting of these receptors. Moreover, the string version demonstrated markedly greater resistance to affibody-mediated inhibition of EGFR endocytosis and degradation, with an IC$_{50}$ of 86.35 nM compared to just 15.85 nM for the knob-into-hole version (Fig. 2k–m). To further quantify these differences, we used the Cheng-Prusoff equation to calculate the $K_D$ values for the EGFR-binding arm in both geometries. The string version displayed a $K_D$ value of 0.92 nM, significantly outperforming the knob-into-hole version, which showed a $K_D$ of 6.31 nM—an ~85% reduction—indicating enhanced binding affinity (Fig. 2n). One possible explanation for the stronger binding (lower $K_D$) observed with the "string" geometry compared to the "knob-into-hole" configuration is the greater conformational flexibility offered by the string format. In this extended arrangement, the EGFR-binding domain has more spatial freedom to adopt an optimal orientation, enabling simultaneous and efficient engagement with both EGFR and HER2 with minimal steric hindrance. This flexibility facilitates tighter and more stable interactions with the receptors, resulting in a lower apparent $K_D$ and improved performance under competitive binding conditions.

To further investigate the impact of geometry and valency on functional activity, we generated an additional construct in the knob-into-hole format containing two EGFR-binding affibodies (termed dual-knob-into-hole) as shown in Supplementary Fig. 4j, k. We then performed a side-by-side comparison of this construct with the string version by evaluating their ability to mediate EGFR endocytosis in the presence of increasing concentrations of an affibody-Fc competitor.

We found that at low competitor concentrations (<25 nM), the dual-knob-into-hole construct modestly enhanced endocytosis. However, at higher competitor concentrations (>25 nM), the string construct consistently outperformed both the single- and dual-valency knob-into-hole formats. These results indicate that valency contributes significantly to endocytosis efficiency, but cannot fully compensate for geometric constraints. In summary, both degrader geometry and valency play important and complementary roles in determining the endocytic capacity of dual-targeting constructs.

This mechanistic study underscores the superior degradation efficiency of the string geometry of FolTAC-dual, confirming that its stronger binding affinity and enhanced endocytosis are achieved through cooperative binding. These findings establish the string version as a more effective configuration for dual receptor degradation.

## FolTAC-mediated POI endocytosis into the lysosome

Previous studies have shown that FRα is constantly endocytosed upon binding with its physiological ligand, folic acid[30,31]. To investigate whether FolTAC exploits this trafficking pathway, we performed exploratory experiments using a pHrodo and chimeric antigen receptor (CAR)-GFP system.

First, we confirmed FRα endocytosis using pHrodo. Notably, upon incubation with a pHrodo-tagged Farletuzumab antibody, cancer cell lines exhibited robust FRα+ endocytosis compared to normal cells (Fig. 3a, b). HeLa cells expressing CAR-GFP on the cell surface were treated with CAR FolTAC, which engaged both the CAR-GFP and FRα, leading to the endocytosis of the CAR-GFP complex (Fig. 3c). Localization analysis using lysosomal markers fused to mCherry demonstrated that CAR-GFP co-localized primarily with the late endosome marker Rab7 and the lysosomal marker Lamp1b, while minimal overlap was observed with the early endosome marker EE1A (Fig. 3c and Supplementary Fig. 5a). These results confirmed that FolTAC-mediated trafficking likely directs the bound proteins into the lysosomal degradation pathway.

## Unraveling the molecular mechanisms of FolTAC-mediated degradation

To further characterize the degradation mechanism of the FolTAC technology, we treated PC9 cells with EGFR FolTAC in the presence of lysosomal inhibitors Chloroquine (CQ) and Bafilomycin (Baf). Both inhibitors blocked EGFR degradation, indicating lysosomal trafficking as the primary degradation pathway (Fig. 3d). Conversely, treatment with the proteasome inhibitor MG132 did not inhibit EGFR degradation, further supporting that FolTAC-induced degradation is lysosome-dependent rather than proteasome-mediated (Fig. 3d and Supplementary Fig. 5b).

To validate whether the degradation is dependent on FRα, several experiments were conducted. Firstly, an FC-Farletuzumab scFv was co-

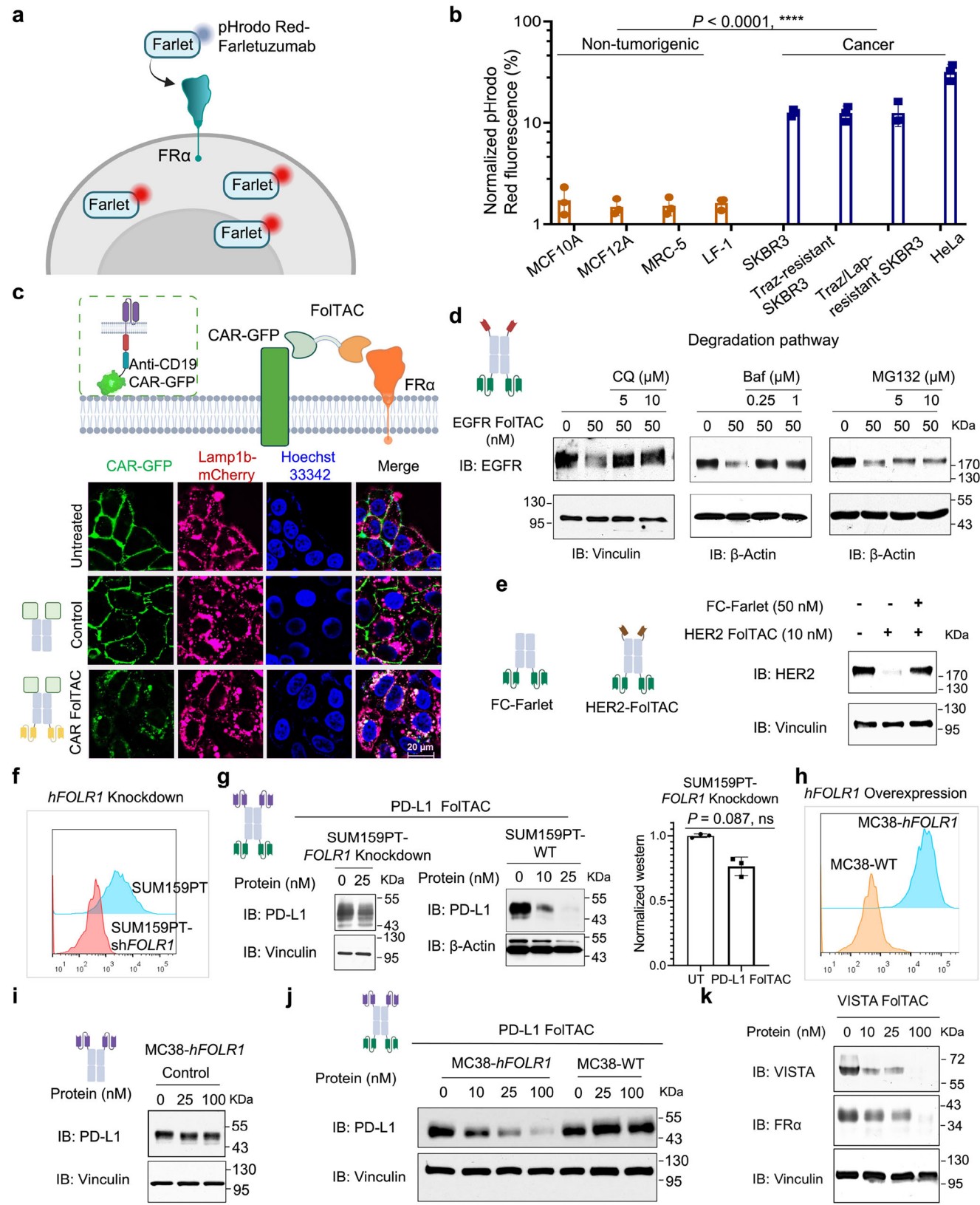

treated with the HER2 FolTAC in SKBR3 cells. The co-treatment induced a competitive inhibition of HER2 degradation by obstructing the binding of HER2 FolTAC to the HER2 receptor (Fig. 3e). Secondly, the IP data in Supplementary Fig. 5c indicate the ternary complex formation between EGFR or HER2 with FRα mediated by FolTAC. Thirdly, SUM159PT cells with *FOLR1* knockdown (KD) were treated

with PD-L1 FolTAC. The results indicated that the KD decreases the PD-L1 degradation efficacy (Fig. 3f, g). Lastly, the mouse cancer cell line MC38, which does not express human FRα, was treated with PD-L1 FolTAC at concentrations up to 100 nM, showing no observable degradation (Fig. 3j). Conversely, engineered cells stably expressing human FRα exhibited gradually increasing PD-L1 degradation with

**Fig. 3 | Molecular mechanism of FolTAC-mediated protein degradation.**
**a** Schematic illustration of Red-Farletuzumab pHrodo to analyze endocytosis comparing cancer versus normal cells. pHrodo is a fluorescent dye with minimal fluorescence at neutral pH but exhibits strong fluorescence in acidic environments following internalization. Both cancer and normal non-tumorigenic cells were incubated with pHrodo Red-Farletuzumab for four hours, and the internalization was assessed using flow cytometry. **b** Flow cytometry analysis of FRα endocytosis. Cancer cells demonstrated a significantly higher capacity to internalize the pHrodo Red-Farletuzumab conjugates compared to normal non-tumorigenic cells. Data are presented as mean ± SD ($n = 3$). The P value was calculated using an unpaired two-sided Student's t-test. **c** Schematics of the strategy to understand the trafficking of CAR-GFP mediated by FolTAC and co-localization imaging of CAR-GFP with lysosomal marker-Lamp1b-mCherry. Cells were treated with PBS, Control, or CAR FolTAC at 100 nM for 24 h before performing the imaging. CAR-GFP (green) marks CAR-expressing cells, Lamp1b-mCherry (magenta) labels lysosomes, and Hoechst 33342 (blue) stains nuclei. Merged images highlight subcellular localization and co-localization patterns. Scale bar: 20 μm (white). **d** Study of the degradation pathway

of FolTAC in PC9 cells using the lysosome inhibitor chloroquine (CQ), bafilomycin (Baf), and the proteasome inhibitor MG132. **e** Competitive rescue of HER2 FolTAC-mediated degradation using FC-Farletuzumab in SKBR3 cells. **f** Flow cytometry analysis of *FOLR1* knockdown SUMP159PT cells. **g** Western blot analysis of PD-L1 degradation following treatment with PD-L1 FolTAC in SUM159PT *FOLR1* knockdown and WT cells. Untreated (0 nM) samples served as the control group for comparison. Quantification analysis of PD-L1 level in SUM159PT *FOLR1* knockdown cells treated with or without FolTAC. "UT" denotes untreated controls. Data represent three independent biological replicates and are presented as mean ± SD. Statistical analysis was performed using an unpaired two-sided Student's t-test, and exact p-values are shown above the figure bars. **h** Flow cytometry analysis of FRα overexpression in MC38 cells. **i, j** mPD-L1 degradation mediated by mPD-L1 control (**i**) in MC38, and FolTAC in MC38-WT and MC38-*hFOLR1* cells (**j**). **k** Western blot analysis of VISTA and FRα co-degradation in MC38 cells overexpressed with FRα and VISTA. Created in BioRender. WANG, Z. (2025) https://BioRender.com/d5n9vj6.

doses of PD-L1 FolTAC ranging from 10 to 100 nM (Fig. 3h–j). To assess whether FolTAC induces FRα degradation, MC38 cells stably expressing both FRα and VISTA were treated with VISTA FolTAC. The results revealed that VISTA FolTAC induced degradation of both VISTA and FRα, therefore highlighting the interplay between target and effector protein degradation (Fig. 3k). These findings collectively validate our hypothesis regarding the FolTAC-mediated degradation mechanism and highlight the significance of FRα-dependent endocytosis in this process. In addition, they emphasize the necessity of meticulous binder screening for designing highly efficient membrane TAC degraders to target Proteins of Interest (POI) to improve their lysosomal trafficking and subsequent degradation.

## Evaluation of the FolTAC-dual platform as a therapeutic strategy for treating Trastuzumab-resistant HER2+ breast cancer

Targeted therapies, such as Trastuzumab, have revolutionized the treatment of HER2+ breast cancer (HER2 + BC)[34,35,70–72]. However, prolonged treatment often leads to resistance, driven by genetic mutations or activation of compensatory pathways such as EGFR signaling[34,35,70–72]. These resistance mechanisms significantly reduce the long-term efficacy of therapies like Trastuzumab (Traz) and Lapatinib (Lap), necessitating alternative strategies to restore drug sensitivity[73]. To overcome this challenge, we developed the FolTAC-dual platform, a modular degrader system designed for the simultaneous degradation of multiple membrane proteins. Using dual targeting of EGFR and HER2 as a representative application, we demonstrate its potential to counteract Traz/Lap resistance in HER2+ cancer models (Fig. 4a and Supplementary Fig. 6a).

To evaluate the impact of FolTAC-dual on HER2+ and EGFR + BC, we treated various drugs on cell viability in both Trastuzumab-sensitive SKBR3-WT and Trat/Lap-resistant SKBR3 cell lines. Resistance was induced by exposing SKBR3-pool2 (Trastuzumab-resistant SKBR3) cells to 0.25 μM Lapatinib[74], creating Traz/Lap-resistant SKBR3 cells[75] (Fig. 4b and Supplementary Fig. 6b). We treated these cell lines with various compounds, including Combo (Carbo/Taxol combination), Lapatinib, Trastuzumab, EGFR/HER2 bispecific affibody-FC, FolTAC-dual degrader v0.6, and FolTAC-dual degrader v1.0 (Fig. 4b and Supplementary Fig. 6). Firstly, the results confirmed that the Traz/Lap resistant SKBR3 cells exhibited altered sensitivity patterns toward Trastuzumab and Lapatinib compared to the SKBR3-WT cells, indicating potential pathways for overcoming drug resistance (Fig. 4b and Supplementary Fig. 6b). Notably, FolTAC-dual degrader v1.0 inhibited proliferation in both SKBR3-WT and Traz/Lap-resistant cells with IC50 values of 10-20 nM (Fig. 4c). Importantly, a cytotoxicity assay across six different normal cell lines confirmed that FolTAC-dual exhibits low toxicity (Fig. 4c and Supplementary Fig. 6c–j). To assess targeting specificity, we performed the coculture assay by adding various

compounds to the coculture system of SKBR3-WT-GFP or Traz/Lap resistant SKBR3-GFP with HFF-1-mCherry. Cells were kept for 6 days and then subjected to fluorescent microscopy imaging to acquire images for fluorescent intensity analysis. Our results revealed that EGFR and HER2 FolTAC-dual v1.0 and Trastuzumab specifically inhibit SKBR3-WT cancer cells while sparing HFF-1 cells (Fig. 4d, e). Notably, the EGFR and HER2 FolTAC-dual v1.0 can inhibit the Traz/Lap resistant SKBR3 cells as expected (Fig. 4e and Supplementary Fig. 7a–e). In contrast, the combination of Carboplatin (Carbo) and Paclitaxel (Taxol) showed high toxicity to the normal cells at three different doses, although exhibiting great inhibiting effects to both SKBR3 cell lines (Fig. 4f–h).

To study the mechanism of EGFR and HER2 FolTAC-dual v1.0 in inhibiting the Traz/Lap resistant SKBR3 cells, we analyzed the downstream of the HER2-AKT signaling pathway. Our results revealed that the FolTAC-dual can inhibit both p-AKT (T308) and p-AKT (S473) sites (Supplementary Fig. 7f), indicating potent inhibition of oncogenic signaling upon treatment. Importantly, this inhibitory effect represents a gain-of-function achieved through the dual degradation capacity of FolTAC-dual, rather than differences in binding affinity or receptor blockade alone. In addition, we confirmed that the affibody binding arm alone is not solely responsible for overcoming resistance. Binding affinity measurements of HER2 affibody-Fc and Trastuzumab in both SKBR3-WT and Traz/Lap-resistant SKBR3 cells revealed no significant difference (Supplementary Fig. 7g), suggesting that the enhanced efficacy of FolTAC-dual may arise from its cooperative configuration or multivalent engagement, rather than from intrinsic differences in binding strength. In summary, the in vitro study by EGFR and HER2 FolTAC-dual demonstrates that it can effectively inhibit downstream AKT oncogenic signaling and diminish receptor-driven survival pathways in Traz/Lap-resistant SKBR3 cells. This highlights the potential of FolTAC-dual as a promising strategy to combat resistance associated with HER2-targeted therapies.

## In vivo evaluation of the EGFR and HER2 FolTAC-dual efficacy

Pharmacokinetic studies were conducted in mice injected with 10 mg/kg FolTAC-dual or control affibody-FC. Blood samples collected over 15 days revealed that FolTAC-dual has a plasma half-life of approximately six days, comparable to the affibody-FC control[26] (Fig. 5a, b). Results showed that the plasma half-life of EGFR and HER2 FolTAC-dual v1.0 is sufficient to exert its therapeutic effects. To further evaluate translational potential, we tested EGFR and HER2 FolTAC-dual in patient-derived organoid (PDO) cultures established from two breast cancer patients (Patient 8 and Patient 10). PDOs treated with 25, 100, and 250 nM FolTAC-dual v1.0 exhibited strong inhibition of cell proliferation, as confirmed by EdU and DAPI staining (Fig. 5c–f and Supplementary Fig. 7h). The dose–response curve characterizing the

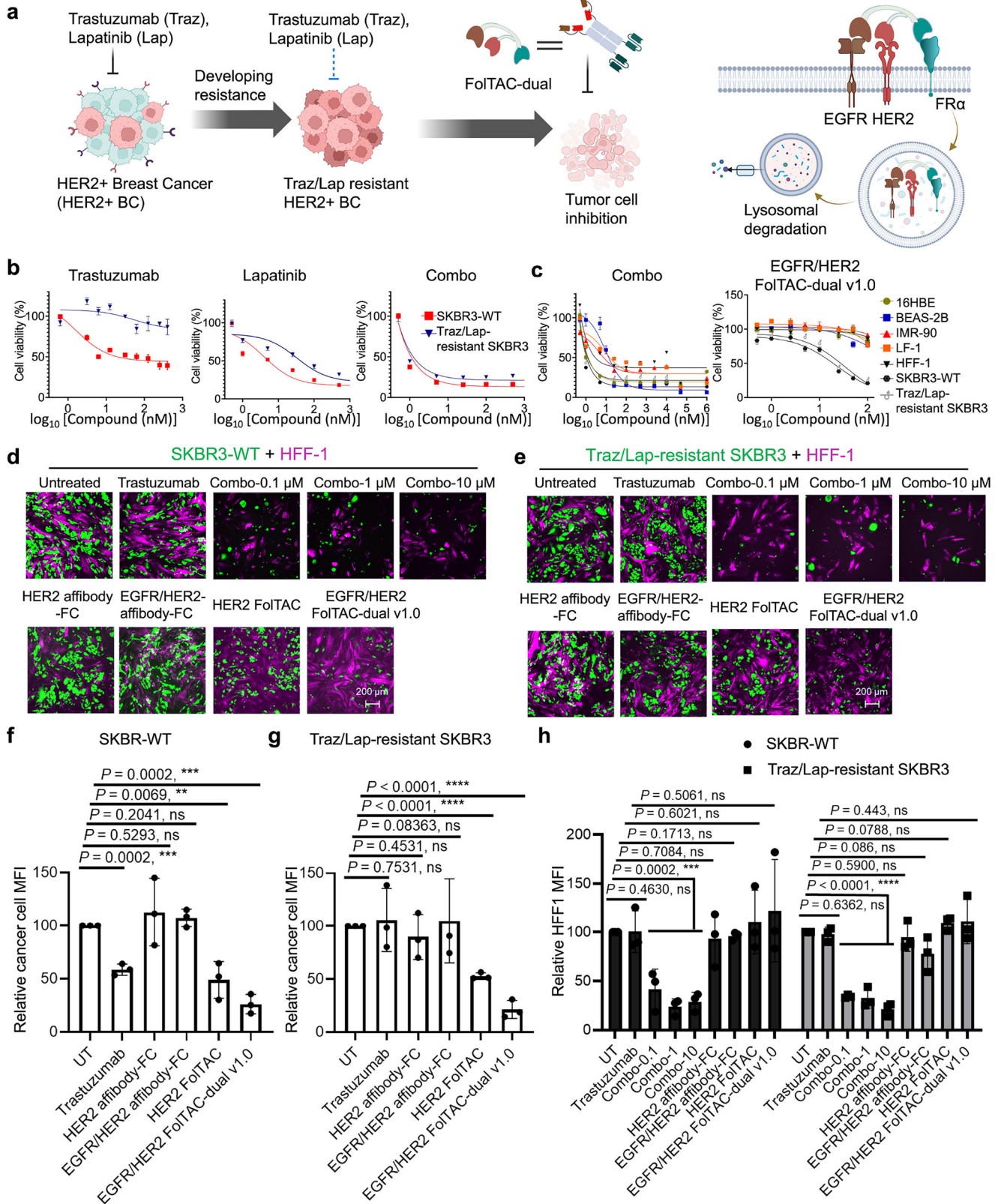

growth inhibition induced by FolTAC-dual was performed using Patient 10 PDOs. The results demonstrate that the EGFR/HER2-targeting FolTAC-dual v1.0 outperforms all control groups in suppressing PDO growth (Supplementary Fig. 7i). Western blot analysis further confirmed significant targeted degradation of EGFR and HER2 at the tested concentrations (Fig. 5g). In addition, the PDO models derived from Patients 8 and 10 exhibit heterogeneous molecular features and compact

structural organization, which may pose barriers to efficient target engagement and limit the observed therapeutic efficacy of the FolTAC-dual strategy. Future studies may assess different dosing regimens or incorporate combination therapies to further elucidate and enhance the therapeutic potential of the FolTAC-dual platform. Together, these findings establish EGFR and HER2 FolTAC-dual v1.0 as a potent and specific treatment for overcoming resistance in HER2+ breast cancer. Its

**Fig. 4 | Development of the FolTAC-dual technology as a strategy to overcome Trastuzumab/Lapatinib resistance. a** Schematic overview of FolTAC-dual therapy for HER2⁺ breast cancer. Left: therapeutic progression in HER2⁺ BC from conventional antibody therapy to dual-targeted degradation. Right: ternary complex formed by FolTAC-dual engaging EGFR, HER2, and FRα to enable selective endocytosis and lysosomal degradation. **b** Cell viability curves of SKBR3-WT and Traz/Lap-resistant SKBR3 cells treated with Trastuzumab, Lapatinib, or Combo (PC, Carbo/Taxol combination). WT cells served as the reference group for comparative profiling against the resistant line. **c** Cell viability of normal human cell lines (16HBE, BEAS-2B, IMR-90, LF-1, HFF-1) and cancer cells (SKBR3-WT and Traz/Lap-resistant SKBR3) following treatment with Combo or EGFR/HER2 FolTAC-dual v1.0. **d**–**e** Co-culture assays evaluating selective anti-cancer activity of FolTAC-dual. SKBR3-WT (**d**) or Traz/Lap-resistant SKBR3 (**e**) cells expressing GFP were co-cultured with HFF-1 fibroblasts expressing mCherry and treated with Trastuzumab, Combo, or EGFR/HER2 FolTAC-dual v1.0 (100 nM for 6 days). SKBR3 cells are shown in green and

HFF-1 fibroblasts in magenta. Cell morphology and spatial distribution were monitored by fluorescence microscopy. Scale bar: 200 μm (white). **f**–**h** Quantification of mean fluorescence intensity (MFI) in co-cultures shown in (**d**–**e**). **f** MFI of SKBR3-WT-GFP and HFF-1-mCherry from (**d**). **g** MFI of Traz/Lap-resistant SKBR3-GFP and HFF-1-mCherry from (**e**). **h** MFI of HFF-1-mCherry from both experiments, assessing off-target toxicity. "UT" denotes untreated control samples. For (**b**–**c**, **f**–**h**), each data point represents the mean ± SD from three independent biological replicates ($n = 3$), each corresponding to a separately seeded and treated well of cells from distinct passages. No technical replicates were used. The unit of study was one well of cell culture. MFIs were quantified using ImageJ software. Statistical comparisons were performed using two-sided unpaired Student's $t$-tests in GraphPad Prism. No corrections for multiple comparisons were applied. Exact $p$-values are shown above bars. Differences were considered statistically significant at $p < 0.05$. Significance annotations: ns, not significant; \*\*$p < 0.01$; \*\*\*$p < 0.001$; \*\*\*\*$p < 0.0001$. Created in BioRender. WANG, Z. (2025) https://BioRender.com/d5n9vj6.

low toxicity in normal cells, coupled with its robust anti-tumor effects in vitro and PDO models, underscores its potential for clinical translation. Further exploration of this platform is warranted to fully realize its therapeutic potential.

Furthermore, the anti-tumor efficacy of EGFR and HER2 FolTAC-dual v1.0 was evaluated in an in vivo model (Fig. 5h). Mice were treated with PBS, Trastuzumab, HER2 FolTAC, a combination of EGFR FolTAC and HER2 FolTAC, or the EGFR/HER2 FolTAC-dual v1.0. Treatment with the FolTAC-dual led to a notable reduction in tumor growth relative to all other groups, as indicated by both excised tumor images and tumor volume measurements (Fig. 5i, j). These findings highlight the therapeutic potential of the FolTAC-dual platform in overcoming resistance to HER2-targeted therapies in vivo.

### Evaluation of the immune-FolTAC platform as an immune-oncology therapy in the MC38 syngeneic model

Co-inhibition therapies targeting PD-L1 and VISTA are emerging as promising strategies in cancer immunotherapy, leveraging their roles in modulating immune responses[76]. While PD-L1 inhibitors re-activate T cells, the interaction between VISTA and PSGL-1 at acidic pH facilitates immune evasion (Supplementary Fig. 8a)[37,57]. Recent studies suggest that dual inhibition of these checkpoints can synergistically enhance immune activation and promote tumor regression, addressing the limitations of single-agent checkpoint therapies, particularly in cancers with high VISTA expression[77,78].

To adapt the FolTAC platform for syngeneic model evaluation, we re-engineered PD-L1 and VISTA FolTAC-dual v1.0 to include a chimeric mouse IgG2A variant suitable for in vivo studies (Supplementary Fig. 8b). This strategy aimed to evaluate the anti-tumor effects of simultaneously degrading PD-L1 and VISTA in an MC38 syngeneic mouse model (Supplementary Fig. 8c). Tumor inoculation experiments were conducted using engineered MC38-h*FOLR1-VSIR* cells in C57BL/6 mice. We assessed in vivo stability, degradation efficiency, and anti-tumor efficacy of the dual degrader (Supplementary Fig. 8d). Pharmacokinetic analyses revealed that PD-L1 and VISTA FolTAC-dual v1.0 exhibits a half-life comparable to EGFR and HER2 FolTAC-dual v1.0 (Supplementary Fig. 8e, f and Fig. 5b). Weight monitoring showed no apparent in vivo toxicity following intraperitoneal (i.p.) injection of 5 mg/kg immune-FolTAC (Supplementary Fig. 8g). We confirmed that human VISTA can be efficiently degraded upon treatment with the immune-FolTAC (Supplementary Fig. 8h, i), consistent with observations in SUM159PT cells (Supplementary Fig. 8i and Fig. 2h), further supporting the platform's versatility across different cellular models. Tumor-bearing mice were treated with PBS, PD-L1 antibody, PD-L1/VISTA bispecific antibody control, PD-L1 FolTAC, or PD-L1/VISTA FolTAC-dual v1.0. Tumor growth analyses showed that FolTAC-dual v1.0 resulted in improved survival rates and greater tumor suppression compared to other treatment groups (Supplementary Fig. 8j, k). Western blot analysis of lysed tumors demonstrated >60%

degradation efficiency in 85% of inoculated tumors treated with FolTAC-dual v1.0 (Supplementary Fig. 8l, m).

The evaluation of the Immune-FolTAC platform in the MC38 syngeneic tumor model (Supplementary Fig. 8n) established its initial practicability and strong potential as an immunotherapeutic strategy for cancer treatment. This approach offers a methodology paradigm and pathway for improving clinical outcomes, particularly in gynecologic and breast cancers with high PD-L1 and VISTA expression, thereby establishing an avenue in immune-oncology therapy.

## Discussion
This study presents the development of the FolTAC-dual to address drug resistance, marking a step forward in the design of dual membrane protein degraders capable of simultaneously targeting two POIs. Our rationally engineered functional modalities—the "knob-into-hole" and "string" configurations—feature distinct spatial arrangements of binding modules, which influence the efficiency of target degradation. Degradation analysis of the target protein revealed that the string configuration exhibited superior degradation efficiency compared to the knob-into-hole design. Mechanistic insights from competitive binding assays further suggested that this enhanced activity stems from improved cooperative binding in the string format.

The EGFR/HER2 FolTAC-dual was evaluated in a Trastuzumab- and Lapatinib-resistant SKBR3 model to investigate its potential in overcoming drug resistance. FolTAC-dual demonstrated robust anti-cancer activity, significantly inhibiting cell proliferation in both wild-type and Traz/Lap-resistant SKBR3 cells. Importantly, compared to traditional chemotherapies that exhibited high toxicity in normal cells, FolTACs revealed markedly reduced toxicity across six normal cell lines, underscoring their potential as a safer therapeutic alternative. Mechanistically, FolTAC-dual exerts a gain-of-function by simultaneously degrading EGFR and HER2, leading to suppression of downstream oncogenic signaling. As part of the proposed mechanism, it inhibits phosphorylation at both p-AKT (T308) and p-AKT (S473), key activation residues within the HER2-AKT survival pathway. In vivo evaluations supported the clinical potential of this approach. Mice xenografted with Traz/Lap-resistant SKBR3 cells treated with FolTAC-dual v1.0 showed greater tumor reduction compared to controls. Pharmacokinetic studies further revealed the degraders' stability and sustained biological activity, supporting their clinical relevance. In conclusion, the FolTAC platform, particularly the dual degrader technology, offers a promising, targeted, and less toxic approach to counteracting resistance in HER2-positive breast cancer, warranting further clinical investigation.

Another major innovation of our research is the development of the immune-FolTAC platform, exemplified by the PD-L1 and VISTA FolTAC-dual v1.0, which represents a significant advancement in dual membrane receptor degrader design. In the MC38 syngeneic mouse model, this dual treatment achieved over 60% degradation efficiency

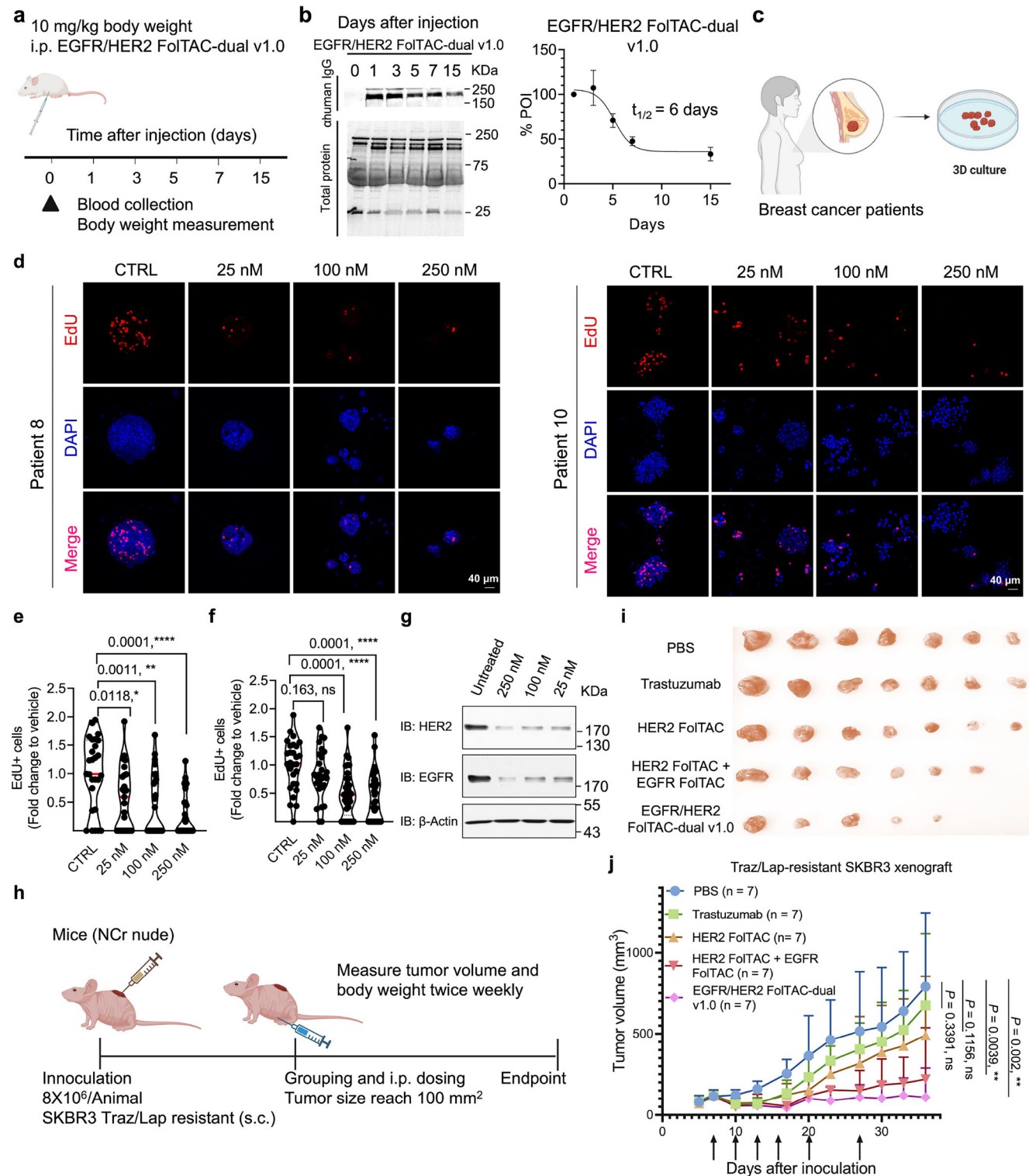

in 85% of the tumors tested, demonstrating its robust degradation ability. However, future in vivo cytotoxicity analyses in blood lineage cells are expected to provide deeper insights into potential systemic effects, further supporting the platform's translational potential. These findings underscore the potential of the immune-FolTAC platform as a transformative approach in immunotherapy, offering a versatile and effective strategy to combat immune resistance and enhance anti-tumor responses in diverse cancer types.

While FolTAC-dual offers significant advantages over traditional therapies such as antibody-mediated blockade and kinase inhibitors,

some limitations warrant further investigation. One notable limitation is the potential for tumors to develop alternative resistance pathways. For example, while FolTAC-dual effectively targets EGFR and HER2 in resistant HER2+ breast cancer, compensatory signaling through other tyrosine kinases or pathways, such as MET or IGF-1R, may emerge over time. In addition, the current preclinical PDO models, while informative, may not fully capture the heterogeneity and complexity of clinical drug resistance, necessitating broader validation across diverse tumor contexts. Similarly, in the context of immune checkpoint degradation, simultaneous targeting of PD-L1 and VISTA may lead to the

**Fig. 5 | Evaluation of EGFR/HER2 FolTAC-dual v1.0 as a therapeutic strategy for Trastuzumab/Lapatinib-resistant breast cancer. a** Schematic of pharmacokinetic studies in mice following intraperitoneal (i.p.) injection of EGFR/HER2 FolTAC-dual v1.0. **b** Western blot analysis of compound levels in mice plasma at defined time points. Band intensities were quantified, and data are presented as mean ± SD with biological repeats ($n = 3$). **c** Diagram of PDO workflow for functional testing. **d** Representative images of EdU (red) and DAPI (blue) staining in PDOs from patients 8 and 10 after 7-day treatment, with the drug refreshed every two days. Magenta marks EdU-positive nuclei. CTRL: control. Scale bar, 40 μm.
**e–f** Quantification of EdU-positive cells in PDOs from patient 8 (**e**) and patient 10 (**f**), treated with vehicle (CTRL), 25, 100, or 250 nM EGFR/HER2 FolTAC-dual v1.0. Each dot represents an individual PDO structure quantified in one biological replicate, and the violin plots represent data distribution from a representative independent experiment out of three repeats. Sample sizes were: CTRL ($n = 27$), 25 nM ($n = 25$),

100 nM ($n = 27$), and 250 nM ($n = 27$) for patient 8; CTRL ($n = 28$), 25 nM ($n = 27$), 100 nM ($n = 33$), and 250 nM ($n = 31$) for patient 10. Data are shown as mean ± SD. Statistical significance was assessed using two-sided unpaired Student's t-tests. Exact $p$-values are indicated above the plots. *, **, **** indicate $p < 0.05$, $p < 0.01$, $p < 0.0001$, respectively; ns, not significant. **g** Western blot analysis showing degradation of EGFR and HER2 in PDOs from patient 8 following treatment. **h** Experimental scheme for in vivo efficacy testing. **i** Representative tumor images from control (PBS, $n = 7$), and treated (Trastuzumab, $n = 7$; HER2 FolTAC, $n = 7$; HER2 FolTAC+EGFR FolTAC, $n = 7$; EGFR/HER2 FolTAC-dual v1.0, $n = 7$) groups at endpoint. **j** Tumor growth in Traz/Lap-resistant SKBR3 xenografts following i.p. injection (5 mg/kg). Two-sided unpaired Student's t-tests were used. Exact p-values are shown above the bars. $p < 0.05$ considered significant; **$p < 0.01$; ns, not significant. Created in BioRender. WANG, Z. (2025) https://BioRender.com/d5n9vj6.

upregulation of other immune checkpoints, such as TIM-3, LAG-3, or CTLA-4, potentially diminishing the long-term efficacy of the therapy. To address these limitations, future studies should explore combination strategies that integrate FolTAC-dual with other therapies, such as inhibitors targeting complementary pathways or immune checkpoint inhibitors like anti-CTLA-4 antibodies. Notably, while neoantigen presentation might be reduced due to degradation, it remains to be evaluated whether the degradation of EGFR into peptides could enhance antigen presentation by MHC-I, representing a potential mechanism for future investigation.

A potential limitation of lysosomal degradation is the loss of neoantigens derived from overexpressed oncogenes, which may reduce MHC-I presentation and impair immunotherapy efficacy. Future strategies could focus on designing selective degradation systems that preserve immunogenic fragments or enhance MHC-I expression. Combining FolTAC-dual with immune adjuvants or checkpoint inhibitors, such as anti-CTLA-4, may further mitigate this effect and enhance anti-tumor immunity. It is also necessary to evaluate the impact of FolTAC-mediated degradation on neoantigen presentation and the tumor immune microenvironment.

In addition, while our strategy demonstrates therapeutic potential in current models, these preliminary findings require further investigation to determine whether the FolTAC approach can effectively overcome drug resistance in broader preclinical settings. Moreover, in future studies, it will be important to expand immunoprofiling and evaluate FolTAC-dual in additional immunotherapy-resistant models to gain a more comprehensive understanding of its therapeutic potential. Future optimization efforts may include incorporating PD-1 or CTLA-4 blockade as combinational therapies, exploring additional preclinical models such as orthotopic tumor models, and refining degrader design to enhance in vivo performance and translational potential. Broader characterization of immune populations, including CD8 + T cells, CD4 + T cells, Tregs, and myeloid cells, alongside cytokine profiling, will help identify pathways that influence treatment efficacy and further validate FolTAC-dual's ability to overcome resistance across diverse tumor contexts.

In conclusion, these findings underscore the potential of the FolTAC-dual platform as a versatile and effective strategy in anti-tumor and cancer immunotherapy. By effectively degrading both EGFR and HER2 as well as PD-L1 and VISTA, this platform provides a potential paradigm in addressing the limitations of single-agent therapies, particularly those associated with long-term resistance. The encouraging results from our established model support further exploration and potential clinical application of this dual degrader approach, especially in gynecologic and breast cancers where FRα, EGFR/HER2, and PD-L1/VISTA are prevalently expressed. The versatility of the FolTAC-dual platform positions it as a promising candidate for next-generation targeted therapies, offering a pathway to develop higher-order degraders and combinatorial treatment strategies for complex diseases beyond cancer.

## Methods

### Ethics statement

All animal experiments were conducted in accordance with institutional guidelines and approved by the Institutional Animal Care and Use Committee (IACUC) at Beth Israel Deaconess Medical Center under protocol number [IACUC #019-2021-24]. Mice were housed in a specific pathogen-free facility under a 12-h light/12-h dark cycle, at an ambient temperature of 20–24 °C and 40–60% humidity, with ad libitum access to standard chow and water. Animals were monitored regularly to minimize pain and distress.

The tissue samples for generating patient-derived organoid models were obtained after receiving written informed consent from all participants, who did not receive compensation. The IRB protocol (#17-627) was reviewed and approved by Dana-Farber/Harvard Cancer Center Scientific Review Committee and Institutional Review Board, and the study abides by the Declaration of Helsinki principles.

### Sex and gender considerations

Only female mice were used in this study due to the HER2+ breast cancer model and to align with the biological characteristics of the patient-derived organoids (PDOs), which were collected from female breast cancer patients. Sex was explicitly considered during study design to ensure biological relevance and reduce variability. Male mice or PDO were not included for these reasons. Sex-disaggregated data were not applicable, as all animals and human samples were female. For patient-derived samples, sex was recorded based on self-report at the time of consent. Additional details are provided in the Nature Portfolio Reporting Summary.

### Plasmid construction

Plasmids were assembled using established techniques in molecular biology. The DNA fragments of the CD19 ectodomain variant, atezolizumab, VSTB112, H7, and Farletzumab scFv, anti-EGFR, and anti-HER2 affibody were synthesized by Integrated DNA Technologies (IDT) and subcloned to a pVitro1 vector for mammalian expression. All FolTACs and controls in this study contain the wild-type IgG1 Fc. The anti-CD19 CAR was generated by fusing a CD19 scFv to the CD8 hinge, CD8 transmembrane, and CD3 Zeta. The mCherry Lamp1, human HER2, human VISTA, and human FRalpha1 plasmids were obtained from Addgene and used directly or subcloned into a pLenti vector for lentivirus production or transient transfection. The GFP and mCherry sequences were PCR amplified from other in-house plasmids and subcloned into a pLenti vector for lentivirus production. The mCherry-tagged Rab7 and LAMP1b expression plasmids were obtained from Addgene (Plasmid #221555 and Plasmid #186576, respectively). Primer sequences used in this study are provided in the Supplementary Data.

### Cell lines

All cell lines were cultured and maintained in 100 mm cell culture dishes at 37 °C with 5% $CO_2$. HEK293T, HFF-1, IMR-90, HeLa, SKOV3,

A549, SUM159PT, HCT116, HCC1806-1, SKBR3, MDA-MB-468, MDA-MB-231, H1299, PC9, LNCAP, BT474, BT549, CAL-120, OVCAR-3, OVCAR-8, MCF-7, HCC1937, and HCC1143 cells were cultured in DMEM medium supplemented with 10% fetal bovine serum (FBS) and 1% penicillin/streptomycin. The cell lines are maintained from Wenyi Wei Lab cell bank (Beth Israel Deaconess Medical Center). 16HBE cells from Ting Wu were cultured in Modified Eagle's Medium (MEM) supplemented with 10% FBS and 1% penicillin/streptomycin. Primary T cells (female) and Primary B cells (female) were cultured in RPMI media supplemented with 10% FBS. The SKBR3-pool2 cell line from Bolin Liu's lab was exposed to 0.25 μM of Lapatinib for 8 weeks to acquire resistance.

## Protein expression
Proteins were expressed and purified from Expi293 cells (Thermo-Fisher Scientific) using a polyethylenimine (PEI) transfection protocol. Enhancers were added 24 h post-transfection. Cells were incubated in a shaking incubator for 5–7 days at 37 °C and 8% $CO_2$. Medium was then collected by centrifugation at $4000 \times g$ for 20 min. Homodimeric FolTACs and controls were purified using Protein A affinity chromatography, and the buffer was exchanged into PBS via spin concentration. Knob-into-Hole FolTACs or controls were purified by Ni-NTA affinity chromatography, with the buffer exchanged into PBS and concentrated. Proteins were kept at 4 °C or flash frozen for storage at −80 °C. The purity and integrity of all proteins were confirmed by SDS-PAGE electrophoresis.

## Lentiviral packaging and stable cell generation
HEK293T cells were cultured until they reached 70–80% confluency, after which the medium was replaced with fresh DMEM supplemented with 15% FBS. A standard Lipofectamine protocol was employed to transfect the cells with the carrier plasmid and lentiviral helper plasmids. The following day, the medium was refreshed with the same DMEM containing 15% FBS. Approximately 36 h post-transfection, the supernatant containing the lentivirus was harvested into 1.5 ml EP tubes and centrifuged at $15,000 \times g$ for 1 min to eliminate cell debris. Target cells were then seeded to achieve 50–70% confluency. The lentivirus was diluted between 1:4 and 1:10 in complete growth media and applied to the cells. The medium was replaced the next day, and the cells were cultured for an additional four days. Stable transductants were subsequently sorted and cultivated for another 5–7 days before being utilized in assays.

## Degradation experiment setups
Cells were seeded and allowed to grow for 6–12 h to a confluency of approximately 60–90%. Then, FolTACs or controls at specified concentrations were added. Cells were incubated with FolTACs or controls for 18–36 h before being harvested for western blotting analysis.

## Flow cytometry
Cells were harvested by centrifugation at $300 \times g$ for 5 min, followed by washing with cold PBS supplemented with 1% BSA and another centrifugation under the same conditions. The cell pellets were then incubated with primary antibodies diluted in PBS + 1% BSA for 10–20 min at 4 °C. After the primary antibody incubation, cells were washed three times with cold PBS + 1% BSA. Secondary antibodies were added, if necessary, and incubated for 20 min at 4 °C. Following the secondary antibody incubation, cells were washed three times with cold PBS + 1% BSA and finally resuspended in cold PBS for flow cytometric analysis. CytExpert software version 2.6 was used for data acquisition, and FlowJo (v 10.8.1) was used for data analysis. The gating strategy employed is presented in Supplementary Fig. 9. The following antibodies were used: APC-conjugated anti-human FOLR1 (Bio-Techne, Clone #548908, Cat. #FAB5646A, 1:200), purified anti-V5-tag antibody

(BioLegend, Cat. #680601, 1:2000), and Alexa Fluor® 647 anti-human IgG Fc (BioLegend, Clone M1310G05, Cat. #410713, 1:1000).

## Western blotting
Cells were lysed using EBC lysis buffer (50 mM Tris pH 7.5, 120 mM NaCl, 0.5% NP-40), supplemented with protease inhibitors from Pierce and phosphatase inhibitors from Calbiochem. Patient-derived organoids (PDOs) were incubated on ice for 1.5 h in Cell Recovery Buffer supplemented with phosphatase inhibitors (Roche #4906845001). PDOs were lysed using RIPA buffer (Boston BioProducts #BP-115) containing protease and phosphatase inhibitors (Thermo Scientific #A32959). Protein concentrations were assessed using the Bio-Rad protein assay reagent on a Beckman Coulter DU-800 spectrophotometer. Approximately 50 μg of whole cell lysates (WCL) were separated by 10% SDS-PAGE at 130 V for 90 min for protein analysis. Proteins were then transferred to PVDF membranes and probed with the indicated specific primary antibodies at 4 °C overnight. After five washes with Tris-buffered saline containing 0.1% Tween-20 (TBST), the membranes were incubated with corresponding secondary antibodies for 1 h at room temperature and washed four times with TBST buffer. The antibodies used included HER2 (#4290S, 1:1000), EGFR (#2232S, 1:1000), anti-PD-L1 (#13684, 1:1000), anti-VISTA (#64953, 1:1000), mouse-VISTA (#54979, 1:1000), anti-CD71 (#13113S, 1:1000), AKT1 (#2938, 1: 1000), p-AKT-T308 (#9275, 1:1000), p-AKT-S473 (#9271, 1:1000), p-mTOR (#5536, 1:1000), and FRα (#34265, 1:1000) from Cell Signaling Technologies; mouse PD-L1 (#ab213480, 1:1000) from Abcam; anti-Actin (A2228, 1:50,000), anti-Vinculin (V-4505, 1:50,000), anti-mouse secondary (A-4416, 1:5000), and anti-rabbit secondary (A-4914, 1:5000) from Sigma; anti-V5-tag Antibody (#680601, 1:1000) from Biolegend. Source data are provided in the Supplementary Information.

## Primary T cell isolation
Human peripheral blood mononuclear cells (PBMCs) were isolated using Ficoll separation in SepMate tubes (STEMCELL Technologies), following the manufacturer's instructions. CD8 + T cells were subsequently isolated from the PBMCs using EasySep Human CD8 + T cell isolation kits (STEMCELL Technologies), in accordance with the provided protocol[79]. The purity of the isolated cells was assessed by flow cytometry with anti-CD3, anti-CD4, and anti-CD8a antibodies (BioLegend). After analysis, the cells were aliquoted, cryopreserved in liquid nitrogen, and stored for future use.

## Primary B cell isolation
Approximately 10 mL of human total blood was placed in a 50 mL Falcon tube, where 750 μL of RosetteSep Human B cell enrichment cocktail (STEMCELL #15024) was added. After gently inverting the tube, the mixture was incubated for 20 min at room temperature. Next, 10 mL of PBS with 2% FBS was added, and the solution was mixed gently. This mixture was carefully layered dropwise over 10 mL of Lymphoprep density gradient medium (STEMCELL #07801) in another 50 mL tube. The tube was centrifuged at $1200 \times g$ for 20 min at room temperature with the brake off. B cells were collected from the interface between the upper plasma layer (red) and the clear Lymphoprep layer below. The isolated B cells were washed twice with 10 mL of 2% FBS in PBS and resuspended in RPMI media supplemented with 10% FBS.

## Fluorescence microscopy
Cells were initially cultured in either Mat-Tek 35-mm glass-bottom Petri dishes or 24-well glass-bottom plates until they reached 30–70% confluency. Subsequently, the growth medium was replaced with fresh growth medium containing either FolTAC or control antibodies. After incubating for 24 h at 37 °C, the cells were washed with PBS and stained with Hoechst 33342 (Thermo Scientific), following the standard

staining protocol provided by BioLegend. Imaging was performed using Nikon Ti inverted, Zeiss LSM880, or KEYENCE BZ-X700 All-in-One. Images were captured using 405-nm, 488-nm, and 647-nm laser settings and subsequently processed with ImageJ.

### Knockdown of *FOLR1* with shRNAs

SUM159PT cells were plated in 6-well cell culture plates 18–24 h prior to transduction to achieve approximately 50% confluency by the time of transfection. Fresh complete DMEM culture media containing 10% FBS and antibiotics were added to each well 30 to 60 min before transfection. The harvested lentiviral supernatant was then introduced to the cells. Assays were conducted on the cells 48 h after shRNA transduction.

### Cell viability experiments

Cells were plated at a density of 1000 to 6000 cells per well in a 96-well tissue culture plate and treated with FolTACs, control antibodies, or small molecule compounds. After a 5–7 day incubation at 37 °C with 5% $CO_2$, Cell Counting Kit-8 (CCK-8, ApexBio, K1018) was added following the manufacturer's instructions. The plates were then incubated at 37 °C for 1–4 h. The spectrophotometric absorbance of the solutions was measured at 470 nm using a CLARIOstar microplate reader (BMG LABTECH).

### Assessing cell proliferation in patient-derived organoid cultures

PDOs were cultured in Matrigel-based 3D domes[80,81] using advanced DMEM/F12 medium supplemented with 1x GlutaMax, 10 mM HEPES, 1% penicillin/streptomycin, B27 supplement (1×, ThermoFisher), 200 ng/ml R-Spondin 3, 5 nM Neuregulin, 80 ng/ml Noggin, 5 ng/ml FGF7, 20 ng/ml FGF10, 5 ng/ml EGF, 10 ng/ml CXCL12, 20 ng/ml IGF-1, 10 ng/ml Osteopontin (all from PeproTech), 500 nM TGFBRII inhibitor A82-01 (Tocris #2939), 500 nM p38 MAPK-inhibitor SB202190 (Selleckchem, #S1077), 500 μM N-Acetylcysteine, 1 mM Nicotinamide, 50 ng/ml hydrocortisone, 0.5 ng/ml 17β-Estradiol, 50 μg/ml Primocin. Medium was refreshed every two days, and organoids were passaged every 7–10 days by mechanical dissociation. To analyze sensitivity to the degrader, PDO fragments were plated into 8-well chamber slides at a density of 200–600 fragments and treated with degrader at concentrations of 25 nM, 100 nM, or 250 nM on day 1, 3, and 5. Cell proliferation was assessed after 7 days by pulsing PDOs with 10 μM 5-ethynyl-2′-deoxyuridine (EdU) for 2 h. PDOs were fixed for 30 min using 4% paraformaldehyde, washed with PBS, and permeabilized with wash buffer (0.3% Triton X-100 in PBS) for 20 min. EdU labeling was performed for 40 min using the EdU Click-IT imaging kit (Invitrogen) according to the manufacturer's instructions. PDOs were washed extensively, incubated for 15 min with DAPI (1 μg/mL), and mounted using Vectashield mounting media containing DAPI (Vector Laboratories). PDOs were imaged with a Zeiss LSM 880 confocal microscope. To assess proliferation, 25-35 PDOs were imaged, and the ratio of EdU-positive cells to the total number of cells was quantified. Statistical analysis was performed with Prism GraphPad, and Student's t-test was used to calculate *p*-values.

### Breast cancer cells/normal cells co-culture assay

SKBR3 and HFF-1 cells were genetically modified to express GFP and mCherry, respectively, via lentiviral transduction, resulting in SKBR3-GFP and HFF-1-mCherry cell lines. These cells were mixed at a 1:2 ratio and seeded at a density of 20,000 cells per well in 12-well plates. The following day, treatments with either FolTAC, control antibodies, or small molecule compounds were initiated and continued for six days. On day 6, fluorescence imaging was conducted using a KEYENCE BZ-X700 All-in-One with a 10x objective. After imaging, the cells were collected and analyzed with FlowJo (v 10.8.1). The relative proportions of SKBR3-GFP to HFF-1-mCherry cells and their mean fluorescent signals under various treatment conditions were quantitatively assessed using ImageJ.

### In vivo FolTACs half-life studies

Nude female nu/nu mice, bred at the Jackson Laboratory and aged 10–12 weeks, were intraperitoneally injected with approximately 10 mg/kg (body weight) of FolTACs or control antibodies. Each experimental condition involved three mice. Blood samples were collected from the lateral saphenous vein using EDTA capillary tubes prior to injection on day 0 and subsequently on days 1, 3, 5, 7, and 15 post-injection. Plasma was collected by centrifuging the blood samples at $2000 \times g$ at 4 °C for 15 min. For Western blot analysis, 1 μL of plasma was diluted in 10 μL of PBS. A Goat Anti-Human IgG antibody (LICOR #926-32232, AB6858) was utilized to detect FolTACs or controls. Band intensities were quantified using ImageJ software. The half-life of the antibodies was calculated using the non-linear regression function in GraphPad Prism.

### EGFR/HER2 FolTAC-dual v1.0 in vivo anti-tumor efficacy

Female Ncr nude mice (NCRNU-F), aged 4 weeks, were obtained from Taconic Biosciences. The right flank of each mouse was inoculated with $8 \times 10^6$ Traz/Lap-resistant SKBR3 cells in 100 μL of PBS containing 50% Matrigel (Corning, 356231). Once the tumor size reached 100 mm² (day 5), the mice were randomly assigned to treatment groups, with 7 animals per group. Each mouse in these groups was then injected with either PBS or HER2/EGFR FolTAC-dual (5 mg/kg body weight) on the indicated days in Fig. 5. Tumor length and width were measured using calipers 2–3 times per week, and tumor volume was calculated using the modified ellipsoidal formula: V = ½ (Length × Width²). When the volume of the largest tumor reached the ethical endpoint (diameter in any direction >2 cm), all mice were euthanized, and the tumors were removed for imaging.

### PD-L1/VISTA FolTAC-dual v1.0 in vivo anti-tumor efficacy

Female C57BL/6J mice (Strain #:000664) at 6 weeks of age were purchased from the Jackson Laboratory. The right flank of each mouse was inoculated with $5 \times 10^5$ MC38-*FOLR1* cells in 100 μL of PBS. Once the tumor size reached a size of 100–200 mm² (days 5–7), mice were randomly assigned to treatment groups, with 15 animals per group. Subsequently, the mice were injected (I.P.) with PBS, PD-L1 antibody, PD-L1/VISTA antibody control-v1.0, PD-L1 Fol-TAC, and PD-L1/VISTA FolTAC-dual v1.0, respectively. Tumor length and width were measured using calipers 2–3 times per week, and tumor volume was calculated using the modified ellipsoidal formula: V = ½ (Length × Width²). When the volume of the tumor reached the ethical endpoint (diameter in any direction > 2 cm), the mice were sacrificed.

### FolTACs in vivo targeted degradation assay

To measure target degradation in tumors, we selected the seven largest tumor samples from the PD-L1/VISTA-FolTAC-dual v1.0 group and four random samples from the PBS groups for analysis. Approximately 10 mg of each tumor sample was sliced and processed in a tissue grinder with the addition of liquid nitrogen. Following this step, 1 mL of RIPA buffer was added to the samples and incubated at 4 °C for 1 h. Subsequently, the samples were sonicated and analyzed using the standard Western blot protocol.

### Transcriptomics analysis across tissues and cancer types

Gene expression data from the Genotype-Tissue Expression (GTEx) and Cancer Genome Atlas (TCGA) were downloaded from the UCSC Xena portal (https://xenabrowser.net/). The expected_count output from RSEM (v1.1.17) was normalized using DESeq2 (v1.44.0) for comparison. A boxplot was created using ggplot2 (v3.5.1).

## Data and statistical analysis

All graphing and statistical analyses were conducted using GraphPad Prism or Microsoft Excel. For all Western assays, an unpaired parametric t-test was performed. For comparisons of gene expression in healthy and cancer cells, a non-parametric Wilcoxon test was chosen to avoid additional assumptions regarding the expression data. A two-tailed $P$ value was used to determine statistical significance in all analyses. A $P$ value of <0.05 was considered statistically significant. ns: $p > 0.05$, *: $p \leq 0.05$, **: $p \leq 0.01$, ***: $p \leq 0.001$, ****: $p \leq 0.0001$.

## Statistics and reproducibility

All experiments were independently repeated at least three times with similar results unless otherwise stated. Figures 1d, e, 3d, e, 3i–k, and 5g present data from a single experiment and were not repeated due to proof-of-concept demonstration or technical constraints. The conclusions drawn from Fig. 1d, e are supported by complementary cell line data in Supplementary Fig. 2 with reproducible results. Similarly, conclusions from Fig. 3c are corroborated by complementary experiments in Supplementary Fig. 5a, conducted with three biological replicates. Figure 3d, e is further supported by consistent findings across additional cell lines, and Fig. 3i–k are reinforced by reproducible results observed at varying dosages. Figure 5g, derived from patient-derived organoids (PDOs), reflects a single experiment due to limited sample availability and technical constraints.

## Reporting summary

Further information on research design is available in the Nature Portfolio Reporting Summary linked to this article.

# Data availability

The datasets generated and/or analyzed during the current study are available upon reasonable request. Source data are provided with this paper.

# Code availability

All data supporting the findings of this study are available within the article and its Supplementary Information files. Transcriptome data from cancer patients and normal tissues were downloaded from the UCSC Xena portal (https://xena.ucsc.edu/). Bioinformatic analysis code for the main and Supplementary Figures has been deposited in Zenodo under the https://doi.org/10.5281/zenodo.16421330 and is openly accessible under the MIT License. The notebook includes methods for accessing the datasets necessary to interpret, verify, and extend the study, along with clear instructions for reuse and reproducibility. No additional restrictions apply.

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

## Acknowledgements

This work was supported in part by the 2023 AACR-AstraZeneca Breast Cancer Research Fellowship (J.H.), The Ludwig Cancer at Harvard Medical School (T.M.), and the NIH grants R35CA253027 (W.W.). Additionally, we thank members of the Wei Lab, Muranen Lab, and Liu Lab for their valuable input to the project and the manuscript.

## Author contributions

Z.W., D.Z., and W.W. conceived the study and designed the research. Z.W. and D.Z. performed the experiments and analyzed the results unless otherwise stated. Z.W., D.Z., and W.W. co-wrote the manuscript. Z.W., D.Z., W.W., H.I., Z.L., J.H., and T.M. co-edited the manuscript. Z.L. and X.S. carried out the transcriptomics analysis. J.H. and T.M. provided input to the human-derived breast cancer organoid research. R.J. provided input to the synergic mouse model design. P.Y., T.H., D.H., Y.Q., and J.W. contributed to the mouse experiments. T.W. provided input to the normal cell line characterization. B.L. provided input to the SKBR3 Trastuzumab/Lapatinib resistant studies. D.Z. and W.W. co-supervised the project. All authors edited and approved the final manuscript version.

## Competing interests

W.W. is a co-founder and stockholder of Rekindle Therapeutics. All other authors declare no competing interests.
