## [Transparent Peer Review file · Nature Communications]

Dual Membrane Receptor Degradation via Folate Receptor Targeting Chimera for Cancer Drug Resistance

Corresponding Author: Dr Wenyi Wei

Version 0:

Reviewer comments:

Reviewer #1

(Remarks to the Author)

This paper presents a dual-targeting fusion protein LYTAC, which utilizes the folate receptor alpha as the lysosome-targeting receptor. The design of a tandem scFv degrader enhances the degradation efficiency of target proteins in vitro. Additionally, the dual-targeting platform induces degradation of HER2/EGFR or PD-L1/VISTA in antibody-resistant models of HER2 or PD-L1, respectively, and elicits anti-tumoral effects.

The authors should cite and discuss the following publication, which is highly relevant to the present work in the LYTAC field, as both studies involve recruitment of the folate receptor.

Zhou, Y.; Li, C.; Chen, X.; Zhao, Y.; Liao, Y.; Huang, P.; Wu, W.; Nieto, N. S.; Li, L.; and Tang, W. Development of Folate Receptor Targeting Chimeras for Cancer Selective Degradation of Extracellular Proteins. *Nat. Commun.* 2024, 15, 8695.

In addition, this reviewer suggests the following revisions.

- Page 7, Lines 264–271

In the comparison between the "string" and "knob-into-hole" configurations, the latter includes one fewer EGFR-binding affibody. The current explanation attributing the observed differences to "conformational flexibility" may not be sufficiently rigorous, especially considering that the EGFR affibody is positioned near the Fc domain in both models. A more detailed structural or mechanistic rationale would strengthen this point.

- Figure 4D–E

These figures present results using only the HER2-FoITAC as a control. Given that the system is designed for dual-targeting, it would be informative to compare the efficacy of the dual-target FoITAC to a combination treatment with both HER2-FoITAC and EGFR-FoITAC. Such a comparison would better illustrate the added value of the dual-targeting design.

- In Vivo Experiments: EGFR/HER2 FoITAC-dual and Immune-FoITAC in MC38 Model

The in vivo studies include only a PBS control group, which limits the ability to interpret the therapeutic effects of the FoITAC-dual constructs. Including additional control groups, as done in the in vitro experiments, would allow for a more comprehensive evaluation of the treatment's efficacy and specificity.

Overall, this study introduces the folate receptor targeting chimera-dual platform, a dual-targeting degrader strategy that effectively overcomes drug resistance by simultaneously degrading membrane proteins such as EGFR/HER2 and PD-L1/VISTA. It demonstrates significant anti-tumor activity, reduced toxicity, and enhanced immune responses in resistant cancer models. The platform shows promise for future clinical applications and combination therapies in cancers with complex resistance mechanisms. This reviewer recommends its publication after addressing the above concerns.

Reviewer #2

(Remarks to the Author)

This manuscript highlights a FoITAC dual platform strategy to address drug resistance. In this platform, Folate Receptor alpha is utilized as an effector for targeted degradation of two proteins (EGFR and HER2 & VISTA and PD-L1). The strategy is based on design and synthesis of engineered antibody scaffolds targeting simultaneously FRA and the two POIs. They demonstrated rational design of their FoITACs, target degradation by Western blot, activity in resistant cell lines and in-vivo activity in mice. The manuscript is clear, well-written and significant to address cancer drug resistance. This manuscript fits within the scope of *Nat. Comm.* and the following revisions are recommended:

- I would like to see a time course of degradation of the conjugates. It appeared that most Western Blots were performed

within 18-36 hours of treatment, but how was this timeframe determined? Is there an optimal time in which degradation is at its max? and related to this question, what is the relevant time to evaluate modulation of the pathway biomarkers (mTOR & AKT) for EGFR degradation?

- Based on the design of their engineered antibody scaffolds and the mechanism of degradation, should Folate Receptor alpha also be degraded? If so, why is this degradation not monitor by western blot as well? And what are the consequences of folate receptor alpha degradation on its own?

- For the mechanistic characterization of the FOLTACs, it would be valuable to demonstrate ternary or quaternary complex formation of the interaction of the conjugate with the FRa, EGFR, HER2 (in cells or biochemically).

- Can the authors elaborate on the differences in amount of degradation between VISTA and PD-L1 in Fig.2 GHI? Since this does not appear to be a catalytic event, similar amounts of POIs should theoretically be degraded, but that is not what is observed by Western blots.

- Figure 1D is not clear whether V0.2 is a CatB linker or not, please clarify the structure.

- Line 258: sentence is grammatically incorrect, please correct.

- Figure 3D: western blot with Baf seems to stabilize/increase expression of EGFR - can the blots be quantified and the authors elaborate if increase expression is indeed the case?

Reviewer #3

(Remarks to the Author)

This study examines a double targeting degrader approach coupled with targeting to the folate receptor to maximize therapeutic index. The engineering is quite elegant and an extensive range of data are presented evaluating different vector designs, mechanisms of action (cell location - lysosome and signaling pathway), and efficacy in cell lines and in tumor models.

For this reviewer, the earlier parts of the manuscript are most convincing, showing the extent of degradation and high inhibition.

Somewhat weaker are the latter PDO/in vivo approaches, where the PK appears favorable, but the antitumor effects appear modest, even disappointing - there is up to two thirds reduction in cell proliferation by the EdU assay (unclear why a conventional cell killing assay was not used as well), and the tumor growth is reduced, but not suppressed. The statement that this then overcomes resistance and is potent is somewhat overstated.

The same is true for the PD-L1-VISTA approach. (Also all the data here are in supplemental). The reduction in tumor growth (SFig 5J-K) appears modest (the Y axis is not labeled in S5J). Again, the claims of efficacy appear overstated.

Thus, there appears to be impressive engineering but whether this approach does provide an approach to overcome appears unclear.

Version 1:

Reviewer comments:

Reviewer #1

(Remarks to the Author)

The authors have addressed all concerns appropriately.

Reviewer #2

(Remarks to the Author)

Upon further review and assessment of the revised manuscript, the authors have addressed all of my concerns and feedback. Great job on this impactful work!

Reviewer #3

(Remarks to the Author)

The authors have responded to my comments satisfactorily.

Point-To-Point Response to Reviewers
(Manuscript ID: NCOMMS-25-12141)

Dual Membrane Receptor Degradation via Folate Receptor Targeting Chimera to Overcome Cancer Drug Resistance

Reviewer #1 (Remarks to the Author):

This paper presents a dual-targeting fusion protein LYTAC, which utilizes the folate receptor alpha as the lysosome-targeting receptor. The design of a tandem scFv degrader enhances the degradation efficiency of target proteins *in vitro*. Additionally, the dual-targeting platform induces degradation of HER2/EGFR or PD-L1/VISTA in antibody-resistant models of HER2 or PD-L1, respectively, and elicits anti-tumoral effects.

Response: We appreciate the reviewer's constructive suggestions to improve our manuscript.

1. The authors should cite and discuss the following publication, which is highly relevant to the present work in the LYTAC field, as both studies involve recruitment of the folate receptor.

Zhou, Y.; Li, C.; Chen, X.; Zhao, Y.; Liao, Y.; Huang, P.; Wu, W.; Nieto, N. S.; Li, L.; and Tang, W. Development of Folate Receptor Targeting Chimeras for Cancer Selective Degradation of Extracellular Proteins. *Nat. Commun.* 2024, 15, 8695.

Response: We deeply appreciate the insightful suggestions from the reviewer, which have significantly contributed to enhancing the quality of our manuscript. We have cited and discussed this important paper in the introduction, Revised Line 94.

In addition, this reviewer suggests the following revisions.

2. Page 7, Lines 264–271

In the comparison between the "string" and "knob-into-hole" configurations, the latter includes one fewer EGFR-binding affibody. The current explanation attributing the observed differences to "conformational flexibility" may not be sufficiently rigorous, especially considering that the EGFR affibody is positioned near the Fc domain in both models. A more detailed structural or mechanistic rationale would strengthen this point.

Response: We thank the reviewer for the constructive comments to improve our manuscript. We agree that a more detailed rationale is needed to clarify the differences in activity between the "string" and "knob-into-hole" constructs. While conformational flexibility remains a contributing factor, we have now expanded our explanation to incorporate valency, structural, and steric considerations. We have added experiments as shown in **Revised Supplementary Fig. 4J and 4K**, and also discussed in Revised Line 278, to provide a more comprehensive structural and mechanistic rationale for the configuration design.

3. Figure 4D–E

These figures present results using only the HER2-FoITAC as a control. Given that the system is designed for dual-targeting, it would be informative to compare the efficacy of the

dual-target FoITAC to a combination treatment with both HER2-FoITAC and EGFR-FoITAC. Such a comparison would better illustrate the added value of the dual-targeting design.

Response: We have repeated this experiment, treated with the combination of both HER2-FoITAC and EGFR-FoITAC within the **Revised Supplementary Fig. 7A and 7B**.

4. *In Vivo* Experiments: EGFR/HER2 FoITAC-dual and Immune-FoITAC in MC38 Model

The *in vivo* studies include only a PBS control group, which limits the ability to interpret the therapeutic effects of the FoITAC-dual constructs. Including additional control groups, as done in the *in vitro* experiments, would allow for a more comprehensive evaluation of the treatment's efficacy and specificity.

Response: As kindly instructed, we have included additional control groups in the *in vivo* experiments, including Trastuzumab, HER2-FoITAC, as well as the combination of EGFR-FoITAC and HER2-FoITAC, as shown in the **Revised Fig. 5I and 5J**. In parallel, we have also updated the patient-derived organoid (PDO) experiments by incorporating the same set of control treatments to ensure consistency across models. These updated data are presented in **Revised Supplementary Fig. 7I**. In addition, the MC38 syngeneic tumor model included a comprehensive set of control groups-PBS, PD-L1 antibody, PD-L1/VISTA antibody control-v1.0, and PD-L1 FoITAC-for direct comparison with the dual-targeting PD-L1/VISTA FoITAC-dual-v1.0.

Overall, this study introduces the folate receptor targeting chimera-dual platform, a dual-targeting degrader strategy that effectively overcomes drug resistance by simultaneously degrading membrane proteins such as EGFR/HER2 and PD-L1/VISTA. It demonstrates significant anti-tumor activity, reduced toxicity, and enhanced immune responses in resistant cancer models. The platform shows promise for future clinical applications and combination therapies in cancers with complex resistance mechanisms. This reviewer recommends its publication after addressing the above concerns.

Response: We sincerely appreciate the reviewer's insightful suggestions to further improve our manuscript.

Reviewer #2 (Remarks to the Author):

This manuscript highlights a FOLTAC dual platform strategy to address drug resistance. In this platform, Folate Receptor alpha is utilized as an effector for targeted degradation of two proteins (EGFR and HER2 & VISTA and PD-L1). The strategy is based on design and synthesis of engineered antibody scaffolds targeting simultaneously FR α and the two POIs. They demonstrated rational design of their FOLTACs, target degradation by Western blot, activity in resistant cell lines and *in-vivo* activity in mice. The manuscript is clear, well-written and significant to address cancer drug resistance. This manuscript fits within the scope of Nat. Comm. and the following revisions are recommended:

Response: We appreciate the insightful suggestions from the reviewer. Following the reviewer's kind guidance, we conducted additional experiments for this round revision, as detailed in point-to-point responses below.

1. I would like to see a time course of degradation of the conjugates. It appeared that most Western Blots were performed within 18-36 hours of treatment, but how was this timeframe determined? Is there an optimal time in which degradation is at its max? and related to this question, what is the relevant time to evaluate modulation of the pathway biomarkers (mTOR & AKT) for EGFR degradation?

Response: As kindly instructed, we performed a time-course experiment, as shown in **Revised Supplementary Fig. 3D**. The newly obtained data indicate that HER2 protein is almost completely degraded between 18 and 24 hours. Based on these results, we selected the 18-36 hours for subsequent analyses. We have also marked this time point in the legends of **Revised Supplementary Fig. 3F and 3G**, where we evaluated the modulation of mTOR and AKT signaling as downstream biomarkers of EGFR degradation.

2. Based on the design of their engineered antibody scaffolds and the mechanism of degradation, should Folate Receptor alpha also be degraded? If so, why is this degradation not monitor by western blot as well? And what are the consequences of folate receptor alpha degradation on its own?

Response: We appreciate the reviewer's insightful suggestions. As kindly suggested, we treated cells with various FOLTAC degraders targeting specific receptors and assessed FR α degradation. The corresponding degradation data have been included in **Revised Supplementary Figs. 3A-3C**. Additionally, we now discuss the downstream consequences of FR α degradation in the revised manuscript (see Revised Line 197). The inserted discussion part is as follows:

"We observed co-degradation events; however, the extent of FR α depletion detected by western blot was influenced by the receptor's baseline expression levels across different cancer cell lines. Notably, cancer cell lines with high FR α abundance exhibited reduced degradation, whereas those with lower FR α expression showed more pronounced depletion, suggesting that FOLTAC activity may preferentially compromise FR α in contexts where its expression is limited. This highlights a potential trade-off between effective target engagement and unintended loss of FR α , depending on receptor expression levels in different cancer contexts."

3. For the mechanistic characterization of the FOLTACs, it would be valuable to demonstrate ternary or quaternary complex formation of the interaction of the conjugate with the FR α , EGFR, HER2 (in cells or biochemically).

Response: Following the reviewer's valuable guidance, we conducted immunoprecipitation (IP) experiments, as shown in **Revised Supplementary Fig. 5C**. The newly obtained results demonstrate ternary complex formation between EGFR/HER2 and Folate Receptor α , mediated by the EGFR/HER2-targeting FoTAC.

4. Can the authors elaborate on the differences in amount of degradation between VISTA and PD-L1 in Fig.2 GHI? Since this does not appear to be a catalytic event, similar amounts of POIs should theoretically be degraded, but that is not what is observed by Western blots.

Response: We appreciate the reviewer's insightful comment regarding the differential degradation levels observed between VISTA and PD-L1 in **Fig. 2G-I**. We have now elaborated on this point in the Revised Line 247. The inserted discussion part is as follows:

"One potential explanation is that differences in binding affinity and expression dynamics of the two POIs may influence degradation efficiency. Notably, the appearance of neo-glycosylated or processed forms of PD-L1 and VISTA (lower bands) adds further complexity, as these species may have differential accessibility to the degradation machinery or altered stability. These variations complicate accurate dose-to-degradation correlations and suggest that substrate-specific factors, including post-translational modifications and receptor trafficking, may contribute to the observed differences in degradation levels."

5. Figure 1D is not clear whether V0.2 is a CatB linker or not, please clarify the structure.

Response: Following the reviewer's kind guidance, we clarified that the structure of v0.2 is without a linker in the Figure and legend of **Fig.1D**.

6. Line 258: sentence is grammatically incorrect, please correct.

Response: Following the reviewer's kind guidance, the grammar of the sentence in Revised Line 272 was corrected. The "and" between "moreover" and "the" has been removed as follows:

"Moreover, the string version demonstrated markedly greater resistance to affibody-mediated inhibition of EGFR endocytosis and degradation, with an IC₅₀ of 86.35 nM compared to just 15.85 nM for the knob-into-hole version (**Fig. 2K, 2L, and 2M**)."

7. Figure 3D: western blot with Baf seems to stabilize/increase expression of EGFR - can the blots be quantified and the authors elaborate if increase expression is indeed the case?

Response: We thank the reviewer for their suggestions. The blots for Baf in **Fig. 3D** were repeated and quantified in **Revised Supplementary Fig.5B**.

Reviewer #3 (Remarks to the Author):

This study examines a double targeting degrader approach coupled with targeting to the folate receptor to maximize therapeutic index. The engineering is quite elegant and an extensive range of data are presented evaluating different vector designs, mechanisms of action (cell location - lysosome and signaling pathway), and efficacy in cell lines and in tumor models.

Response: We sincerely thank the reviewer for recognizing the elegance of our engineering approach and the comprehensive nature of our mechanistic and efficacy evaluations. We appreciate the acknowledgment of our efforts in designing and characterizing the dual-targeting degrader system across molecular, cellular, and animal model levels.

For this reviewer, the earlier parts of the manuscript are most convincing, showing the extent of degradation and high inhibition.

Response: We are pleased that the reviewer finds the early-stage mechanistic and cellular efficacy data convincing. These results underscore the robustness of our platform in achieving selective and efficient degradation through folate receptor-mediated targeting. We have also thoroughly revised the manuscript in response to the reviewers' comments, including the addition of new assays, careful text editing, and reorganization of data to avoid overstatement.

1. Somewhat weaker are the latter PDO/*in vivo* approaches, where the PK appears favorable, but the antitumor effects appear modest, even disappointing - there is up to two thirds reduction in cell proliferation by the EdU assay (unclear why a conventional cell killing assay was not used as well), and the tumor growth is reduced, but not suppressed. The statement that this then overcomes resistance and is potent is somewhat overstated.

Response: We appreciate this important feedback and agree that additional clarification and experimental data are warranted to better support our claims regarding *in vivo* efficacy. In the revised manuscript, we have taken the following steps to address the raised concerns:

1. **New assay inclusion:** To directly address the reviewer's suggestion, we have included conventional cell viability assays (IC₅₀ assay) in patient-derived organoid (PDO) model Patient 10 to complement the EdU proliferation data in **Revised Supplementary Fig. 7I**. Due to the characteristics of patient 8 PDOs, we cannot dissociate the PDOs into single cells, thus we could not perform the cell viability assay with this limited sample. Since PDOs grow slower than conventional *in vitro* models and require higher doses/more frequent dosing, we have repeated the experiments with more frequent dosing of the degrader over 7 days (refreshing the drugs every two days). Apart from IC₅₀ analysis, fluorescent imaging of EdU- and DAPI-stained PDOs from Patients 8 and 10 was repeated following a 7-day treatment with EGFR/HER2 FoITAC-dual v1.0, during which the drug was refreshed every two days. As shown in **Revised Fig. 5D**, EGFR/HER2 FoITAC-dual v1.0 demonstrates enhanced inhibitory efficacy compared to the single-treatment condition shown in our previous data (**Revised Supplementary Fig. 7H**). Besides, we repeated the *in vivo* data, including more

controls, such as single degraders and HER2 antibody, to compare with dual degrader as shown in the **Revised Fig. 5I and 5J**. To improve therapeutic efficacy, we included additional drug injections compared to our previous experiment.

2. Clarified language: We revised our claims in the manuscript to more accurately reflect the level of antitumor effect observed, emphasizing that our approach leads to partial suppression of tumor growth and improved target degradation, rather than complete tumor regression. The “overcome” is removed in the title and manuscript to avoid overstating.
3. Model limitation discussion: We have added a discussion (Revised Line 461) to the manuscript regarding the challenges and limitations of achieving strong tumor suppression in aggressive or late-stage tumor models, and highlight that our current design serves as a platform for future optimization. This limitation of PDO model is also discussed in the Results section (Revised Line 388) as follows:

“Additionally, the PDO models derived from Patients 8 and 10 exhibit heterogeneous molecular features and compact structural organization, which may pose barriers to efficient target engagement and limit the observed therapeutic efficacy of the FoITAC-dual strategy. Future studies may assess different dosing regimens, or incorporate combination therapies to further elucidate and enhance the therapeutic potential of the FoITAC-dual platform.”

2. The same is true for the PD-L1-VISTA approach. (Also all the data here are in supplemental). The reduction in tumor growth (SFig 5J-K) appears modest (the Y axis is not labeled in SF5J). Again, the claims of efficacy appear overstated.

Thus, there appears to be impressive engineering but whether this approach does provide an approach to overcome appears unclear.

Response: We thank the reviewer for pointing this important issue out. Following the reviewer’s comments, we have revised the manuscript in the following aspects:

1. Figure improvement: We have corrected the missing Y-axis label in **Revised Supplementary Fig. 8J** to facilitate interpretation.
2. Text revision: As with the primary study, we have revised the description of PD-L1-VISTA efficacy to avoid overstating the impact and instead focus on the observed partial tumor inhibition.
3. Discussion: We have included a brief note in the discussion (Revised Line 481) highlighting the potential of PD-L1 and VISTA dual degradation as a platform for immune modulation, while acknowledging the current limitations in efficacy. In the discussion, we refine our claims regarding resistance to note that our approach shows potential but requires further validation in genetically or pharmacologically resistant models to draw definitive conclusions. The inserted discussion part is as follows:

“Additionally, while our strategy demonstrates therapeutic potential in current models, these preliminary findings require further investigation to determine whether the FoITAC approach can effectively overcome drug resistance in broader preclinical settings. Moreover, in future studies, it will be important to expand immunoprofiling and evaluate FoITAC-dual in additional immunotherapy-resistant models to gain a more comprehensive understanding of its therapeutic potential. Future optimization efforts may include incorporating PD-1 or CTLA-4 blockade as combinational

therapies, exploring additional preclinical models such as orthotopic tumor models, and refining degrader design to enhance *in vivo* performance and translational potential.”